



# French coastal network for carbonate system monitoring: The CocoriCO₂ dataset

Sébastien Petton[1], Fabrice Pernet[1], Valérian Le Roy[1], Matthias Huber[1], Sophie Martin[2], Éric Macé[2], Yann Bozec[2], Stéphane Loisel[2], Peggy Rimmelin-Maury[3], Émilie Grossteffan[3], Michel Repecaud[4], Loïc Quemener[4], Michael Retho[5], Soazig Manac'h[5], Mathias Papin[6], Philippe Pineau[7], Thomas Lacoue-Labarthe[7], Jonathan Deborde[8], Louis Costes[8], Pierre Polsenaere[8], Loïc Rigouin[9], Jérémy Benhamou[9], Laure Gouriou[9], Joséphine Lequeux[10], Nathalie Labourdette[11], Nicolas Savoye[11], Grégory Messiaen[12], Elodie Foucault[12], Vincent Ouisse[12], Marion Richard[12], Franck Lagarde[12], Florian Voron[13], Valentin Kempf[13], Sébastien Mas[13], Léa Giannecchini[13], Francesca Vidussi[14], Behzad Mostajir[14], Yann Leredde[15], Samir Alliouane[16], Jean-Pierre Gattuso[16,17], Frédéric Gazeau[16]

[1]Ifremer, Univ Brest, CNRS, IRD, LEMAR, F-29840 Argenton, France
[2]Adaptation et Diversité en Milieu Marin, AD2M Station Biologique de Roscoff, CNRS, 29680 Roscoff, France
[3]Institut Universitaire Européen de la Mer (OSU-IUEM), Univ Brest, CNRS-UAR3113, F-29280, Plouzané, France.
[4]Ifremer Centre de Brest REM/RDT/DCM, 29280 Plouzané, France
[5]Ifremer, Laboratoire Environnement et Ressources du Morbihan et Pays de Loire, 56100 Lorient, France
[6]Ifremer, EMMA, F-85230 Bouin, France
[7]Littoral Environnement et Sociétés, LIENS Université de la Rochelle, CNRS, 17000 La Rochelle, France
[8]Ifremer, Laboratoire Environnement et Ressources des Pertuis Charentais, 17390 La Tremblade, France
[9]Ifremer, Laboratoire Environnement et Ressources d'Arcachon, 33120 Arcachon, France
[10]URA POREA/OASU, CNRS, Université de Bordeaux, La Rochelle Université, INRAE, 33600 Pessac, France
[11]UMR EPOC/OASU, Université de Bordeaux, CNRS, Bordeaux INP, 33600 Pessac, France
[12]MARBEC, Univ Montpellier, CNRS, Ifremer, IRD, Sète, France
[13]OSU OREME, CNRS, Univ Montpellier, IRD, IRSTEA, 34200 Sète, France
[14]MARBEC, Univ Montpellier, CNRS, Ifremer, IRD, Montpellier, France
[15]Géosciences Montpellier, CNRS, Univ Montpellier, 34000 Montpellier, France
[16]Sorbonne Université, CNRS, Laboratoire d'Océanographie de Villefranche, F-06230 Villefranche-sur-Mer, France
[17]Institute for Sustainable Development and International Relations, Sciences Po, F-75007 Paris, France

*Correspondence to*: Sébastien Petton (sebastien.petton@ifremer.fr)

**Abstract.** Since the beginning of the industrial revolution, atmospheric carbon dioxide (CO₂) concentrations have risen steadily and have induced a decrease of the averaged surface ocean pH by 0.1 units, corresponding to an increase in ocean acidity of about 30%. In addition to ocean warming, ocean acidification poses a tremendous challenge to some marine organisms, especially calcifiers. The need for long-term oceanic observations of pH and temperature is a key element to assess the vulnerability of marine communities and ecosystems to these pressures. Nearshore productive environments, where a large majority of shellfish farming activities are conducted, are known to present pH levels as well as amplitudes of daily and seasonal variations that are much larger than those observed in the open ocean. Yet, to date, there are very few coastal observation sites where these parameters are measured simultaneously and at high frequency.



To bridge this gap, an observation network was initiated in 2021 in the framework of the CocoriCO₂ project. Six sites were selected along the French Atlantic and Mediterranean coastlines based on their importance in terms of shellfish production and the presence of high- and low-frequency monitoring activities. At each site, autonomous pH sensors were deployed both inside

and outside shellfish production areas, next to high-frequency CTD (conductivity-temperature-depth) probes operated through two operating monitoring networks. pH sensors were set to an acquisition rate of 15 min and discrete seawater samples were collected biweekly in order to control the quality of pH data (laboratory spectrophotometric measurements) as well as to measure total alkalinity and dissolved inorganic carbon concentrations for full characterization of the carbonate system. While this network has been up and running for more than two years, the acquired dataset has already revealed important differences

in terms of pH variations between monitored sites related to the influence of diverse processes (freshwater inputs, tides, temperature, biological processes). Data are available at https://doi.org/10.17882/96982 (Petton et al., 2023a).

## 1 Introduction

In the last centuries, the industrial revolution has resulted in a continuous increase in the concentration of carbon dioxide ($CO_2$) in the atmosphere (IPCC, 2023). A significant part of this anthropogenic $CO_2$ (26%) has been absorbed by the ocean

(Friedlingstein et al., 2022), resulting in a decline of the global average surface pH by 0.1 units. The decline in seawater pH and associated carbonate ion concentrations, a process referred to as Ocean Acidification (OA; Caldeira and Wickett, 2003), has potential impacts on marine organisms, particularly those that rely on calcification. Indeed, marine calcifiers, such as shallow-water tropical corals, shellfish, and calcareous plankton, are examples of ecologically and economically important organisms that may be affected by OA in the coming decades. (Gazeau et al., 2013; Gattuso et al., 2014; Lutier et al., 2022).

The carbonate chemistry of the ocean surface layer is subject to notable temporal and spatial variations that arise from a combination of local physical conditions (*e.g.* temperature, salinity), biological processes (*e.g.* primary production, dissolution and precipitation of calcium carbonate), basin-scale circulation patterns, and exchanges between the ocean, land, and atmosphere. The resulting changes in pH, carbonate ion concentrations and saturation states with respect to calcium carbonate minerals have important implications for ecosystem function and global biogeochemical cycling. To assess the future effects

of OA on marine organisms in the coming decades, there is a growing need, driven by both scientific and societal demands, for high quality data on the status and trends of OA worldwide. This is evidenced by various initiatives such as the United Nations Sustainable Development Goal (UN SDG) 14.3.1, which calls for annual reporting on global seawater pH observations.

Coastal areas, where a large majority of shellfish farming activities take place, are particularly important to monitor because

of their high biological productivity and economic significance. In these areas, pH levels and variations differ from those observed in the open ocean (Takeshita et al., 2015; Fassbender et al., 2016; Feely et al., 2016). Long-term high-resolution observations that capture diurnal to seasonal variations are critical for characterizing natural variability and understanding carbonate parameters (Rosenau et al., 2021; Sutton et al., 2022), indicating a critical need for increasing monitoring efforts in





nearshore environments. This knowledge is crucial to develop a comprehensive understanding of the impacts of OA on marine

ecosystems and to develop effective management strategies to mitigate the adverse effects on these areas (Riebesell and Gattuso, 2015). Moreover, fixed time-series observations also play a unique role in ocean monitoring, as they can support multidisciplinary observations and process studies and serve as high-quality reference stations for validating satellite measurements and Earth system models (Kwiatkowski et al., 2020). These observations can also be used as critical climate records if they are of sufficient duration and measurement quality to detect the anthropogenic signal above the natural

variability of the ocean carbon system.

Over the French coastline, two study sites benefit from rather long historical records of ocean carbonate chemistry. Kapsenberg et al. (2017) showed, thanks to a discrete (weekly) sampling strategy, that the rate of pH change was -0.0028 ± 0.0003 units pH yr$^{-1}$ in a coastal Mediterranean site (Bay of Villefranche-sur-Mer) between 2007 and 2015. Gac et al. (2020) also provided a robust multi-annual dataset from 2015 to 2019 based on hourly measurements deployed on a buoy in the southern Western

English Channel. They demonstrated that the tidal transport of coastal and offshore water masses strongly impacted the daily and seasonal variability of the partial pressure of $CO_2$ ($pCO_2$), although biological productivity was the main driver of $pCO_2$ dynamics with large inter-annual variability.

Generally, rapid fluctuations of pH in coastal areas due to processes mentioned above frequently largely surpass the long-term pH trend. Consequently, the signal indicating the acidification of coastal seawater caused by anthropogenic $CO_2$ uptake can

be partially masked and less discernible if the sampling strategy is not adapted. High-frequency pH measurements are therefore needed to properly understand both the current state and the underlying mechanisms that drive short-term pH variations as well as better evaluating interannual variability.

In the present paper, we introduce a new monitoring network of the carbonate chemistry that includes high-frequency pH measurements and low-frequency measurements of dissolved inorganic carbon and total alkalinity. The CocoriCO$_2$ network

was initiated in 2021 at six nearshore areas. It includes multiple monitoring stations equipped with pH probes and conductivity-temperature-depth (CTD) sensors and a similar discrete sampling strategy between stations, which enable uniformed data acquisition and processing. We demonstrate the capabilities of this network by analyzing the temporal and spatial variability of pH, total alkalinity, and dissolved inorganic carbon in these coastal systems. The network provides a deeper understanding of OA in typical coastal ecosystem and the impact of pH variations on the oyster vulnerability.

## 95  2 Material and Methods

### 2.1 Study sites and observation strategy

The objective of the network relies on monitoring carbonate parameters along the French coastal systems. Six different areas have been chosen based on the presence of oyster farming activity and existing monitoring efforts to facilitate maintenance of the sensors and seawater sampling on a regular basis (see following sections). In each area, we aimed at monitoring one site



located next to oyster farms and another one outside shellfish production areas more subjected to oceanic conditions. Four oceanic sites have been selected among the national program COAST-HF (Coastal Ocean Observing System High-frequency, https://www.coast-hf.fr). It gathers 14 French coastal stations with essential oceanographic parameters at a sub-hourly frequency: temperature ($\pm$ 0.1 °C), conductivity ($\pm$ 0.3 mS cm$^{-1}$), *in vivo* fluorescence ($\pm$ 10%), and turbidity ($\pm$ 10%). In contrast, nearly all farming sites are part of the Ifremer observatory network ECOSCOPA (https://ecoscopa-

donnees.ifremer.fr). Within this network, temperature ($\pm$ 0.1 °C) and salinity ($\pm$ 0.5) have been recorded at a 15 min frequency since 2010. In the frame of these two networks, all sites are visited on a bi-monthly to monthly basis allowing to download recorded pH$_T$ data (pH on total scale), clean sensors and perform low-frequency samplings (see 2.2.2).

Except for the Bay of Morlaix, the chosen study sites also belong to two existing low-frequency environmental networks (Fig. 1): The SOMLIT (Service d'Observation en Milieu LITtoral, INSU-CNRS; https://www.somlit.fr/) and the Ifremer REPHY

(Observation and Surveillance Network for Phytoplankton and Hydrology in coastal water; Belin et al., 2021) which have been contributing to long-term time series for more than 20 years (Lheureux et al., 2023) and providing crucial parameters (*e.g.* nutrient concentrations) necessary for data processing (see §2.3.1).

All monitoring sites are shown in Fig. 1 and a brief description of each area is given hereafter following a latitudinal gradient from North to South. The Ifremer Bouin oyster growth facility has been chosen as an additional site because it has a significant

interest to monitor seawater pH$_T$ evolution during the early stages of oyster growth, as it is a critical period for their development. The data obtained from this site will provide us with a deeper understanding of the impact of pH$_T$ levels on the growth of oysters and their capacity to survive in their natural habitat.



**Figure 1: Localization of the high-frequency monitoring sites (blue dots) from the CocoriCO₂ network. The pink and green dots indicate SOMLIT and REPHY low-frequency stations, respectively, from which nutrient data were acquired.**

### 2.1.1 Bloscon and Figuier sites in the Bay of Morlaix

The Bay of Morlaix is located in the south of the Western English Channel and forms a wide indentation in the north of Finistère. The hydrodynamics is governed by a strong semi-diurnal tidal cycle whose maximal range reaches 8 m. The bathymetry is characterized by shallow areas (roughly 10 m depth) and intertidal flats. It is influenced by continental freshwater inputs from two small rivers, the Morlaix and the Penzé rivers, with an average flow of 4.3 m$^3$ s$^{-1}$ and 2.8 m$^3$ s$^{-1}$, respectively (Gac et al., 2020).



The oceanic site is located at the north cardinal buoy Basse de Bloscon (48°43'43.3''N - 3°58'09.7''W) next to the Bloscon harbor. The probes are fixed at 4 m depth onto a stainless-steel frame attached to the mooring. The farming site is set next to the isolated hazard mark Figuier (48°40'28.1''N - 3°56'09.3''W) in the Penzé strait. The probes are fixed at 50 cm above the
seabed onto a stainless-steel frame.

As these two sites were not part of pre-existing monitoring networks, we do not have access to any historical data that could provide us with valuable insights.

### 2.1.2 MAREL and SMART sites in the Bay of Brest

The Bay of Brest is a semi-enclosed macrotidal system located at the western end of Brittany. It is connected to the Iroise Sea
through a 1.8 km wide by 6 km long and roughly 50 m deep inlet (called the Goulet de Brest). It undergoes strong currents due to its complex geometry and topography and is characterized by a dominant semi-diurnal tide whose maximal range reaches 7.3 m (Petton et al., 2020). As its average depth is only 8 m, the back-and-forth flow at each tide prevents stratification nearly everywhere (Le Pape and Menesguen, 1997). The hydrology is dominated by freshwater runoffs coming mostly from the Aulne, Elorn, and Mignonne rivers, with respectively average winter flows of 54, 10, and 3 m$^3$ s$^{-1}$ respectively (Frère et
al., 2017; Petton et al., 2023b).

Located at the entrance of the Bay of Brest, the oceanic site corresponds to the COAST-HF MAREL-Iroise buoy (48°21'28.7''N - 4°33'05.4''W) which records data every 20 min at 2 m depth since 2000 (Rimmelin-Maury et al., 2020). Next to the Mignonne river mouth, the farming site is located over a native flat oyster bed on the COAST-HF SMART-Daoulas buoy (48°19'06.4"N - 4°20'08.2"W) where parameters are monitored 50 cm over the seabed at a 15 min frequency since 2016
(Petton et al., 2021).

### 2.1.3 MOLIT and Men-Er-Roué sites in the Mor Braz area

This area encompasses the Quiberon Bay on the west and the Vilaine Bay on the east in the southern part of Brittany. The whole area is characterized by weak semi-diurnal tidal currents, and therefore subjected to stratification processes. The Vilaine Bay is a shallow area (15 m) directly influenced by the Vilaine and Loire rivers with an average flow of 70 m$^3$ s$^{-1}$ and 850 m$^3$
s$^{-1}$, respectively (Lazure et al., 2009). The Vilaine Bay has undergone eutrophication for several decades mainly due to the high nutrient inputs from the Vilaine and Loire Rivers (Ratmaya et al., 2019). Quiberon Bay is also a shallow area (10 m) exposed to the indirect freshwater inputs of the Loire and Vilaine river plumes which generally spread on a northwestward course along the coast, particularly during early spring (Lazure and Jegou, 1998). The weak vertical mixing causes strong haline stratification (Planque et al., 2004). During thermal stratification in spring and summer, northwestern winds may induce
local upwelling (Lazure and Jegou, 1998; Puillat et al., 2004).

Located at the river mouth of the Vilaine river, the oceanic site is set at the COAST-HF MOLIT-Vilaine buoy (47°27'36.4''N - 2°39'23.7''W) which has recorded hourly data since 2011 at surface and bottom thanks to a pumping system (Retho et al.,



2022). The pH probe is deployed independently at surface level and records data at a 15 min frequency. The farming site is located next to flat oyster farming fields, where probes are deployed over the former ECOSCOPA Men-Er-Roué site

(47°32'19.4''N - 3°05'24.7''W) in the western part of Quiberon Bay and record data at 50 cm over the seabed at a 15 min frequency.

### 2.1.4 Rochelle and D'Agnas site in the Marennes-Oléron Basin

The Marennes-Oléron Basin is the marine region of the Charente-Maritime in Southwestern France. It is a semi-closed shallow basin with a mean depth of 8 m. Characterized by an average tidal range of 5 m and the presence of more than 58% of intertidal

areas, the basin hosts the largest oyster production in France, covering one third of its intertidal areas (Goulletquer and Le Moine, 2002). The basin is mainly connected to the Atlantic Ocean in the north by the Pertuis d'Antioche, a strait contained between two islands (Ile de Ré in the north and Ile d'Oléron in the south-west). The hydrology is dominated by freshwater runoffs coming from the Charente and the Seudre rivers with an average flow of 51 and 3 $m^3$ $s^{-1}$, respectively (Soletchnik et al., 1998).

Initially, the oceanic site should have been deployed in the center of Pertuis d'Antioche. However, due to the intense marine traffic and professional fishing activities, getting the authorization for a mooring buoy was delayed. It was then decided to deploy the pH probe in the La Rochelle cargo harbor (46°09'33.7''N - 1°14'21.5''W) at 3m depth with a 15 min acquisition frequency. Among oyster farming fields, probes are deployed at sub-surface level with a 15 min acquisition frequency on the ECOSCOPA D'Agnas buoy at the southern part of the basin (46°09'33.7''N 1°14'21.5''W).

### 2.1.5 BOUEE-13 and Le Tès sites in the Bay of Arcachon

The Bay of Arcachon is a semi-enclosed lagoon on the southwestern coast of France connected to the Atlantic Ocean. It has a distinctive triangular shape that encompasses a complex network of channels and intertidal flats in the inner Bay (Cayocca, 2001). Intense tidal exchange occurs at the interface between the ocean and the Bay whereas the hydrodynamic exchange is weak over the intertidal areas. The tide cycle is semidiurnal whose maximum range reaches 4.6 m and the mean tidal prism

constitutes two thirds of the inner volume (Plus et al., 2009). The main freshwater runoff comes from the Leyre river with a mean annual flow of 17.1 $m^3$ $s^{-1}$. The Bay is an important area for oyster farming and characterized as the first place for oyster's larvae recruitment in France.

Located in the inlet, the oceanic site is set at the COAST-HF BOUEE-13 (44°37'58.8''N - 1°13'58.8''W), a fairway marine buoy which has recorded data since 2017 at a 10 min frequency at sub-surface level. The probes are fixed onto a stainless-steel

frame attached to the buoy itself. Situated among oyster farming fields, probes are deployed in the study site ECOSCOPA Le Tès (44°39'53.4''N - 1°07'51.5''W) in the eastern part of the Bay and record data at 50 cm over the seabed at a 15 min frequency.



### 2.1.6 Thau lagoon and offshore Sète in the Mediterranean sea

The Thau lagoon is located on the Mediterranean coast of France and is characterized by a total area of 75 km$^2$ and an average
depth of 4 m (Fiandrino et al., 2017). The lagoon has two inlets, namely the Sète channel to the east (responsible for 90% of
the water exchange with the Mediterranean Sea) and the Grau de Pisse-Saumes inlet to the west, which connects it to the sea.
Due to the low amount of water coming from rivers and hydro-morphological profile, the process of water renewal in the
lagoon is slow, taking around 133 d (Fiandrino et al., 2017). The hydrodynamics of the Gulf of Lion continental shelf located
offshore is primarily influenced by wind. The farming site is located on a shellfish farming structures, and probes are deployed
in the experimental breeding fixed table of the study site ECOSCOPA Marseillan (43°22'44.4''N - 3°34'17.4''E), in the
western part of the lagoon. The data were recorded at a fixed 2 m depth below the surface at a 15 min frequency since 2012.
Located to the south east of Thau Lagoon, the oceanic site is set at the COAST-HF BESSète (Bottom Experimental Station of
Sète - 43°19'36.1''N - 3°39'42.0''E) which initially recorded currents and wave data to characterize storm impacts (Michaud
et al., 2013). A subsurface buoy was then deployed in 2017 to survey coastal river plumes with a 15 min record frequency.

### 2.1.7 Ifremer Bouin facility

As one task of the CocoriCO$_2$ project targets to determine the ocean acidification's effect over spat oyster, this site is equipped
to allow carbonate system estimation over the largest area of spat production in France. Environmental monitoring is set at the
Ifremer Bouin Marine Mollusks Platform (46°57'52.7''N - 2°02'41.0''W) located next to the Bay of Bourgneuf in Vendée. It
provides Ifremer laboratories with pre-grown marine mollusks for scheduled experiments in research projects. The probes are
deployed at the inlet of the seawater pumping circulation system and record data at a 15 min frequency.

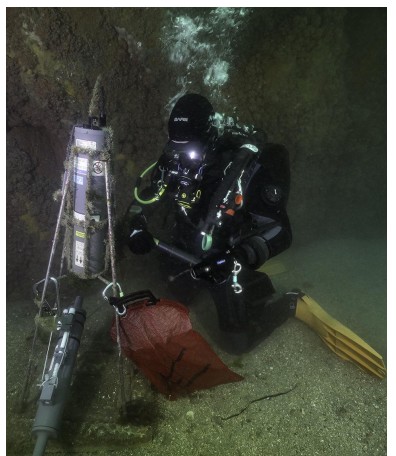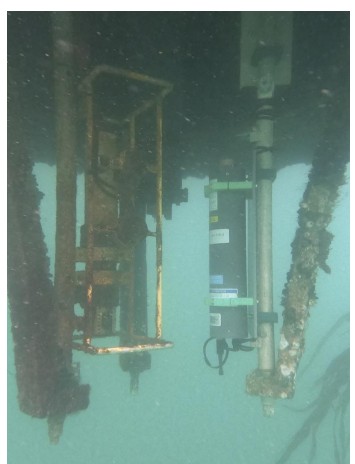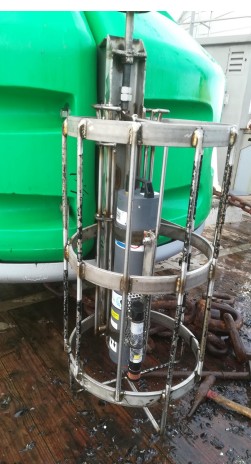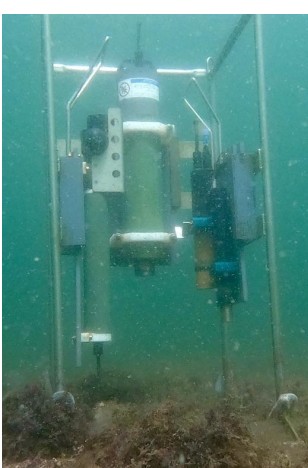

**Figure 2: Pictures of different setups from left to right: submarine site Figuier in the Bay of Morlaix (W. Thomas / CNRS), subsurface mooring COAST-HF MAREL Iroise in the Bay of Brest (S. Petton / Ifremer), subsurface buoy in ECOSCOPA D'Agnas site in the Marennes-Oléron Basin (J. Deborde / Ifremer) and submarine site ECOSCOPA Men-Er-Roué in the Mor Braz (S. Petton / Ifremer).**



## 2.2 Instrument deployments, sampling and analytical methods

In order to assess the whole carbonate system, two of the following parameters have to be recorded: pH on the total scale ($pH_T$), $pCO_2$, total alkalinity ($A_T$) and/or dissolved inorganic carbon ($C_T$). Temperature, salinity and pressure (in case of deep acquisition) as well as nutrient concentrations are also mandatory. The pair of $A_T$ and $C_T$ is usually considered in marine environments to reduce the overall uncertainties of the propagated parameters (Orr et al., 2018). Currently, a significant challenge lies in the absence of available *in situ* instruments that can measure $A_T$ or $C_T$ at high frequency. By combining high-frequency $pH_T$ and $pCO_2$ sensors that are commercially available, it is theoretically possible to compute the whole carbonate system. However, in addition to inducing important purchase and maintenance costs, using this couple of variables leads to important propagated errors in the calculations of the other carbonate variables (Gattuso et al., 2014). Instead, as high-frequency salinity data are available at all study sites, it is possible to reconstruct high-frequency $A_T$ values from salinity-$A_T$ relationships obtained for each site and based on discrete (bi-weekly) sampling for both parameters. Although this workaround may require additional computation and processing, the assessment of carbonate chemistry dynamics based on $pH_T$-$A_T$ offers an interesting solution for obtaining accurate data in the absence of $A_T$ or $C_T$ high-frequency sensors.

### 2.2.1 High-frequency monitoring

When the project started, only a few high-frequency probes were available to record either $pH_T$ or $pCO_2$. They relied on either spectrophotometric measurements (SAMI-pH or SAMI-pCO2, Sunburst Sensors©) or potentiometric sensors (SeaFET V2 pH, Sea-Bird Scientific©). As the filtration system used by SAMI sensors is subjected to clogging by suspended matter and as we already had experience with SeaFET V1 (Satlantic©), the SeaFET V2 pH was chosen for monitoring $pH_T$ at high frequency for this network. Principally based on an Ion Sensitive Field Effect Transistor (ISFET) sensor, it combines a commercially available pH DuraFET electrode (Honeywell©) and two different reference electrodes (Martz et al., 2010), referred to as internal and external. The key element is that the external electrode is sensitive to salinity and more stable that the internal one (Bresnahan et al., 2014).

Each SeaFET of the network is deployed in an autonomous mode next to a CTD probe which is part of the following list: SMATCH MP3 or MP6 (NKE©), WiMo (NKE©), STPS (NKE©), WiSens CTD (NKE©), SBE37-SM (Sea-Bird Scientific©) and SBE37-SMP-ODO (Sea-Bird Electronics©). The acquisition frequency at each site depends on the local environmental conditions but is generally 15 min, except for COAST-HF MAREL Iroise (20 min) and COAST-HF ARCACHON Bouée 13 (10 min). A summary of the sensors used at each site, their deployment depth, acquisition frequency, the starting date of $pH_T$ measurements and the DOI associated with the publicly available datasets is provided in Table 1. Pictures of different setups are given in Fig. 2. On most sites, instruments are deployed on a sub-surface mooring mode thanks to pre-existing navigation channels or oceanographic buoys. For three of them (Figuier, COAST-HF SMART Daoulas, ECOSCOPA Men-Er-Roué), SeaFET and CTD probes are deployed on benthic metallic structure with sensors at 50 cm above seabed (Table 1).



The DuraFET component of SeaFET sensors needs to be replaced at a frequency which varies upon the chemical composition of seawater, generally one year. Another drawback of this sensor is its time adaptation when deploying the probe in a new site. It could reach up to five days for the internal electrode and more than 15 days for the external one. To avoid gaps in datasets, a pre-conditioning of the sensors is done with filtered seawater taken in the field site prior to the deployment. All the technical

field maintenance (sensors cleaning, data upload, SeaFETs swap) is recorded in an Excel monitoring sheet for each site. The general recommendations that framed this network are detailed in Bresnahan et al. (2014).

### 2.2.2 Sensor maintenance and seawater sampling

Maintenance as well as *in situ* calibration/validation are required regularly to achieve a good quality standard for carbonate chemistry, especially in coastal areas where biofouling processes might deteriorate the signal of high-frequency probes. Only

mounted with a copper guard, the SeaFET V2 pH is not sold with an active biofouling cleaning system. Each site is visited on a bi-monthly to monthly basis (mainly depending on meteorological conditions) in the frame of the COAST-HF and ECOCOSPA monitoring efforts. During these visits, before cleaning the SeaFET sensor for removing biofouling, and uploading data, a discrete sampling is realized next to the probe within 5 min of the programmed acquisition time. This time synchronization accuracy is needed especially for the Atlantic sites due to macrotidal forcing. Sampling is performed with

Niskin bottles and seawater is stored in two replicate 500 mL borosilicate bottles following the Standard Operating Procedure 1 of Dickson et al. (2007). A volume of 200 µL of saturated mercuric chloride ($HgCl_2$) is added when the sample is not analyzed within three hours in order to limit biological activity, and bottles are sealed with Apiezon L grease. These samples are used to determine *in situ* $pH_T$ with an accurate spectrophotometric method in order to correct high-frequency $pH_T$ signals (see §2.3.2) and for additional $A_T$ and $C_T$ analyses (see §2.2.3).

Depending on instrumentation resources of the different partners, CTD casts are performed next to the field site with other probes to check for the accuracy of temperature and salinity of the long-term deployment. These discrete measurements are saved in an Excel monitoring sheet.

### 2.2.3 Laboratory analyses

The first sampling bottle is used for determining *in situ* $pH_T$ with a spectrophotometric method and the use of purified m-cresol

purple (purchased from Robert H. Byrne's laboratory, University of South Florida), according to Standard Operating Procedure 6b of Dickson et al. (2007). The dye solution is prepared in high purity deionized water at a concentration of 2 mmol $L^{-1}$. Whenever possible, the analysis is performed within a few hours to limit biological activity which could degrade measurement precision. For each sample, a minimum of three repeatability $pH_T$ measurements is realized. To check for the accuracy of spectrophotometric measurements, a Tris buffer (provided by Andrew Dickson's laboratory, Scripps University, USA or the

French National Metrology Laboratory) is used on a regular basis and each laboratory participates in annual inter-comparison exercises (Capitaine et al, submitted). Whatever the laboratory temperature measurement is, each final result is expressed at



25 °C ($pH_T^{25°C}$) thanks to a numerical correction based on $A_T$ (initial estimation of the carbonate system at laboratory temperature followed by a conversion from the spectrophotometric measurement temperature to 25 °C) and saved in an Excel monitoring sheet.

The second sampling bottle is stored in the dark, at 4 °C. Then, batches of bottles collected within six months are sent to the SNAPO-CO₂ (Service National d'Analyse des Paramètres Océaniques du CO₂) at Sorbonne University in Paris, France. This laboratory is specialized in $A_T$ and $C_T$ analyses with potentiometric titration based on that of Edmond (1970) and uses the program of Dickson and Goyet (1994) for the calculation of the equivalent points. Its precision expressed by the standard deviation of triplicate measurements is less than 3 μmol kg⁻¹ for both parameters, and its accuracy is always compared to

seawater certified reference material purchased from Andrew Dickson's laboratory (Scripps University, USA). The analysis results for each bottle are eventually kept depending on the SNAPO-CO₂ quality flag (2: correct / 3: doubtful)



**Table 1: Localization of the sites monitored in the CocoriCO₂ network. Additional deployment information is also indicated. Low-frequency (bi-montlhy) nutrient data were obtained from nearby monitoring stations as part of the SOMLIT (Service d'Observation en Milieu LITtoral, INSU-CNRS) and the REPHY (Observation and Surveillance Network for Phytoplankton and Hydrology in coastal water) networks.**


| Region | Name | Position | Instruments | Depth | pH$_T$ monitoring | Nutrients | Official DOI |
|---|---|---|---|---|---|---|---|
| Bay of Morlaix | Figuier | 48°40'28.1" N 3°56'09.3" W | WiSens CTDS SeaFET pH | 50 cm above seabed 15 m below Chart Datum | Mar 2021 15 min | REPHY Pont de la Corde | |
| | Bloscon | 48°43'43.3'' N 3°58'09.7'' W | WiSens CTDS SeaFET pH | Subsurface (4 m) | Mar 2021 15 min | SOMLIT Estacade | |
| Bay of Brest | COAST-HF SMART Daoulas | 48°19'06.4'' N 4°20'08.2'' W | SBE37-SMP-ODO SeaFET pH | 50 cm above seabed 3 m below Chart Datum | Mar 2017 15 min | REPHY ETUDE Pointe du Château | 10.17882/86020 |
| | COAST-HF MAREL Iroise | 48°21'28.7'' N 4°33'05.4'' W | SMATCH MP6 SeaFET pH | Subsurface (2 m) | Nov 2020 20 min | SOMLIT Portzic | 10.17882/74004 |
| Mor Braz | ECOSCOPA Men-Er-Roué | 47°32'19.4'' N 3°05'24.7'' W | SBE37-SMP-ODO SeaFET pH | 50 cm above seabed 3 m below Chart Datum | Nov 2020 15 min | REPHY Men-Er-Roué | |
| | COAST-HF MOLIT Vilaine | 47°27'36.4'' N 2°39'23.7'' W | SMATCH MP6 SeaFET pH | Subsurface (2 m) | Jan 2021 15 min | REPHY Ouest Loscolo | 10.17882/46529 |
| Marennes Oléron Basin | ECOSCOPA D'Agnas | 45°52'10.1'' N 1°10'25.7'' W | WiSens CTDS SeaFET pH | Subsurface (2 m) | Feb 2021 15 min | REPHY Auger | 10.17882/86131 |
| | La Rochelle | 46°09'33.7'' N 1°14'21.5'' W | SBE37-SM SeaFET pH | Subsurface (2 m) | Jan 2022 15 min | SOMLIT Antioche | |
| Bay of Arcachon | ECOSCOPA Le Tès | 44°39'53.4'' N 1°07'51.5'' W | WiSens CTDS SeaFET pH | Subsurface (2 m) | Jan 2021 15 min | REPHY ARCH Le Tès | 10.17882/86131 |
| | COAST-HF Bouée 13 | 44°37'58.8'' N 1°13'58.8'' W | SMATCH MP6 SeaFET pH | Subsurface (2m) | Jan 2021 10 min | SOMLIT Bouée 13 | |
| Thau lagoon and offshore Sète | ECOSCOPA Marseillan | 43°22'44.4'' N 3°34'17.4'' E | SMATCH MP3 SeaFET pH | Subsurface (2 m) | Nov 2020 15 min | REPHY Marseillan | 10.17882/86131 |
| | COAST-HF BESSète | 43°19'36.1'' N 3°39'42.0'' E | WiMo plus SeaFET pH | Subsurface (3 m) | Nov 2020 15 min | SOMLIT Sète | |
| Ifremer Bouin oyster growth facility | | 46°57'52.7'' N 2°02'41.0'' W | WiSens CTDS SeaFET pH | Incoming flux | Jan 2021 15 min | - | |



### 2.3 Data processing

#### 2.3.1 Temperature, salinity and nutrient databases

For COAST-HF stations, high-frequency temperature and salinity are recovered in different ways as this network has not a
unified data qualification process. The first step consists in gathering files of different sources, official CORIOLIS database
(https://www.coriolis.eu.org) and local files from additional spare probes, into a single one for each station. Then, based on
low-frequency measurements available within REPHY and SOMLIT networks, doubtful data are identified by flagging them.
This last step mainly concerns salinity data. For ECOSCOPA sites and both sites located in the Bay of Morlaix, a similar
approach is realized where the data acquisition relies on alternating deployment of instruments for short periods between 15
and 30 d. CTD probes are regularly calibrated either at the Ifremer Metrology Laboratory or directly at the manufacturer
throughout annual maintenance. The Bouin facility processes their data and only transmits validated data.

Total concentrations of silicate, phosphorus and ammonium are estimated upon samples taken at the nearest sampling point
from either REPHY or SOMLIT networks. First, monthly climatological means are estimated for each site based on the overall
dataset and a daily climatology is created. Then, as the frequency of these data varies upon sites, nutrients are linearly
interpolated on high-frequency dataset with a threshold of 30 d. If there is more than 30 d between two consecutive nutrient
samplings, the daily climatology estimation is used to fill the gap.

#### 2.3.2 Correction of the high-frequency pH$_T$ signal

The post-processing of the pH SeaFET data is directly based on discrete measurements of pH$_T$ using the spectrophotometric
method (see 2.2.3) and has been well documented since the integration of the DuraFET sensor in oceanic probes (Bresnahan
et al., 2014; Martz et al., 2015; McLaughlin et al., 2017). Briefly, the operation consists in estimating the reference potential
$E_0^*$ of both internal and external electrodes at a fixed temperature (25 °C). First, temperature and salinity from the external
CTD is linearly interpolated at SeaFET data acquisition timestep without temporal extrapolation. If no CTD data are available
within a 6 h interval, the internal DuraFET temperature values are used and salinity is set to missing (which will result in no
coherent external pH signal).
Then, the combination of corresponding raw voltage signals and spectrophotometric pH$_T$ data is used over a specific
deployment period. This period should correspond to the timespan when the SeaFET is deployed. However, it could be
shortened by the operator and depends on the *in situ* calibration primarily controlled by the biofouling development. If an
offset appears between the SeaFET and the discrete sample, an in-depth cleaning of both sensors is required as specified by
the manufacturer. This physical, sometimes chemical, operation may modify the reference potentials and therefore induce a
new deployment number. In case of no available discrete sample, the SeaBird factory coefficients are used to estimate pH$_T$
values. This last case is rare in our dataset and happened only on a few occasions, especially at the early stage of the network.
The deployment matching between SeaFET data and discrete pH$_T$ values is saved in the Excel monitoring sheet.



The last step involves the choice of the most accurate electrode according to *in situ* calibration. A first check is completed by focusing on the difference between both electrodes which generally is the sign of biofouling development. Afterwards it is

quite simple to select the best signal over a deployment period. A last manual data cleaning is performed by attributing a quality flag to each $pH_T$ measurement (1: good data / 2: probably good data / 3: probably poor data / 4: bad data / 9: missing value) following Ocean Data Standards (IOC, 2013).

### 2.3.3 Low-frequency carbonate chemistry

Based on low-frequency (bi-monthly) $A_T$ and $C_T$ measurements as well as temperature, salinity and nutrient data, computation

of carbonate chemistry variables is performed thanks to Python scripts based on Pandas library (McKinney, 2010) with an import of PyCO2SYS package v1.8.1 (Humphreys et al., 2022). According to the community's current best practices (Dickson et al., 2007; Orr et al., 2018), the following constants are used: carbonic acid dissociation constants $K_1$ and $K_2$ from Lueker et al. (2000), fluoride dissociation constant $K_F$ from Perez and Fraga (1987), and bisulfate dissociation $K_S$ from Dickson (1990). Total boron concentration is estimated from salinity using the global ratio of Lee et al. (2010). Among the output variables,

computed $pH_T$ data ($pH_{Comp}$) are compared to direct $pH_T$ measurements using the spectrophotometric method ($pH_{Spectro}$).

### 2.3.4 High-frequency carbonate chemistry

To obtain high-frequency $A_T$ evolution, multiple regression models are used to examine the relationship between $A_T$ and temperature, salinity, ammonium, phosphorus and silicate for each site upon discrete samplings. For each model, explanatory variables are selected based on the Bayesian information criteria (BIC) in a stepwise method. This means that explanatory

variables were added into the model one by one by selecting at each step the one that minimizes the BIC criteria. These statistical analyses are conducted using the GLMSELECT procedures of the SAS software package (SAS 9.3, SAS institute, Carry, USA).

## 3 Results

### 3.1 Data availability and quality assessments

### 3.1.1 Data availability

Despite the COVID-19 crisis, all sites were instrumented in early 2021 (see Table 1). Table 2 gives the ratio of available and qualified data for each site from 1st January 2021 until 31st December 2022. For the sites in the Bay of Morlaix, the installation of structures on pre-existing channel buoys delayed the start of monitoring until March 2021 (10% data lost). Similarly, for the site offshore La Rochelle, authorization requests proved to be complex due to various maritime concerns (ferry and cargo

navigation, trawler fishing zone, etc.), resulting in a one-year delay (50% data lost). An additional constraint of coastal environmental monitoring is the professional and recreational maritime traffic. The two buoys in the Bay of Arcachon have




directly experienced these issues, with several collision events causing, in the worst cases, disruptions to the moorings and destruction of the high-frequency sensors.

Otherwise, the main issue leading to data invalidation remains the development of biological fouling. Since the SeaFET is
only equipped with a passive protective system using a perforated copper cap, damages in both electrode signals were observed during the first 15 days and a bi-monthly manual cleaning frequency was not sufficient. The rate of data loss demonstrates random fluctuations depending on sites and encountered environmental conditions, with a variability that differs from year to year. In order to alleviate these biofouling problems, the SeapHOx (Sea-Bird©) system (SeaFET connected to an CTD SBE37) was tested in the Bay of Quiberon farming, but its pumping system clogs during events of slightly turbid water. Nevertheless,
it is worth noting that ratios of available and qualified data are comparable to those of temperature and salinity measurements in the different sites.

**Table 2: Percentage of available and validated high-frequency data for each station between 1ˢᵗ January 2021 and 31ˢᵗ December 2022**

| | Bay of Morlaix | | Bay of Brest | | Mor Braz | | Marennes Oléron Basin | | Bay of Arcachon | | Thau lagoon and offshore Sète | | Ifremer Bouin |
| | Figuier | Bloscon | COAST-HF SMART Daoulas | COAST-HF MAREL Iroise | ECOSCOPA Men-Er-Roué | COAST-HF MOLIT Vilaine | ECOSCOPA D' Agnas | La Rochelle | ECOSCOPA Le Tès | COAST-HF Bouée 13 | ECOSCOPA Marseillan | COAST-HF BESSète | |
|---|---|---|---|---|---|---|---|---|---|---|---|---|---|
| Temperature | 89.4 | 91.2 | 99.8 | 98.8 | 91.4 | 94.0 | 95.2 | 48.8 | 89.6 | 70.9 | 99.7 | 84.3 | 99.8 |
| Salinity | 82.2 | 73.7 | 98.6 | 92.8 | 91.4 | 84.0 | 94.3 | 35.4 | 84.7 | 70.9 | 97.9 | 58.7 | 99.8 |
| pH | 86.9 | 90.6 | 96.7 | 88.9 | 82.7 | 92.1 | 85.7 | 48.6 | 86.0 | 65.6 | 96.2 | 83.0 | 90.9 |

**3.1.2 Low-frequency data quality**

Following the Global Ocean Acidification Observing Network (GOA-ON; http://www.goa-on.org/) requirements and governance plan (Newton et al., 2015), measurements of carbonate chemistry parameters must be associated with maximal uncertainties in order to reach a "weather" or a "climate" goal. The former implies an uncertainty of ~10 μmol kg⁻¹ in $A_T$ and $C_T$ and 0.02 in pH$_T$. It is defined as a set of measurements of a quality that is sufficient to identify relative spatial patterns and
short-term variations, supporting mechanistic responses to the impact on local, immediate ocean acidification dynamics. The "climate" goal is more ambitious and is defined as a set of measurements of a quality that is enough to assess long-term trends



with a defined level of confidence, supporting detection of the long-term anthropogenically driven changes. It implies an uncertainty of ~2 µmol kg$^{-1}$ in measurements of $A_T$ and $C_T$ (~0.1%) and 0.003 units in pH$_T$.

Direct pH$_T$ measurements using the spectrophotometric method provided results associated with uncertainties ranging between 0.001 and 0.01 (minimum of three analytical replicates), depending on the laboratory in charge of the measurements and the period of analysis. Results of $A_T$ and $C_T$ measurements from the SNAPO-CO$_2$ and flagged as "2: correct" are below the uncertainty level of 0.15%. Based on these considerations, our collected data meet the "weather" quality objective.

Following Orr et al. (2018), the average uncertainties of the derived carbonate parameters are estimated with PyCO2SYS using forward finite-differences for the discrete samplings. With overall accuracies of 0.1 °C and 0.5 for temperature and salinity

(COAST-HF and ECOSCOPA validated accuracy), respectively, as well as 4 µmol kg$^{-1}$ for silicate, 0.1 µmol kg$^{-1}$ for phosphorus and 0.1 µmol kg$^{-1}$ for ammonia (provided by REPHY and SOMLIT networks), the uncertainties associated with the use of the low-frequency $A_T$ - $C_T$ pair (uncertainties of 2 µmol kg$^{-1}$ for both variables) are $2.75 \times 10^{-10}$ mol H$^+$ (about 0.013 units pH$_T$), 15 µatm $p$CO$_2$, 0.12 unit for the aragonite saturation state, and 0.18 for the calcite saturation state. Propagation of uncertainties was performed using the standard uncertainties in the equilibrium constants and total borate concentrations (Orr

et al., 2018). Again, the propagated uncertainties for pH$_T$ meet the "weather" quality objective.

Another way to check for the quality low frequency carbonate chemistry assessments is to compare pH$_T$ analyzed using the spectrophotometric method (pH$_{Spectro}$), and computed from $A_T$ - $C_T$ measurements (pH$_{Comp}$), both based on synchronized sampling on each site. Figure 3 shows these comparisons and the linear relationships established from aggregated farming and oceanic sites at each site. Analyzed and computed pH$_T$ data from the Bay of Brest, Mor Braz and Mediterranean sites exhibit

very good agreements (r$^2$ > 0.98, p < 0.001). In contrast, samples from the Bay of Morlaix present poor correlation with large differences that can reach up to 0.1 pH$_T$ unit. The laboratory involved in this site has also directly measured $A_T$ within two weeks pending sampling in order to compare with measurements obtained from the SNAPO-CO$_2$. The difference between these measurements ranges between -41.9 and 48.7 µmol kg$^{-1}$. These proven inconsistencies highlight conservation issues due to a too long storage period (more than one year for all samples), however this issue is still under investigation. Samples from

the last two regions (Marennes-Oléron basin and Bay of Arcachon) reveal differences only for specific dates. Because of its random nature, we clearly suspect a discrepancy in numerical estimation due to the lack of consideration of organic alkalinity in these mesotrophic sites but, again, this requires more investigation.



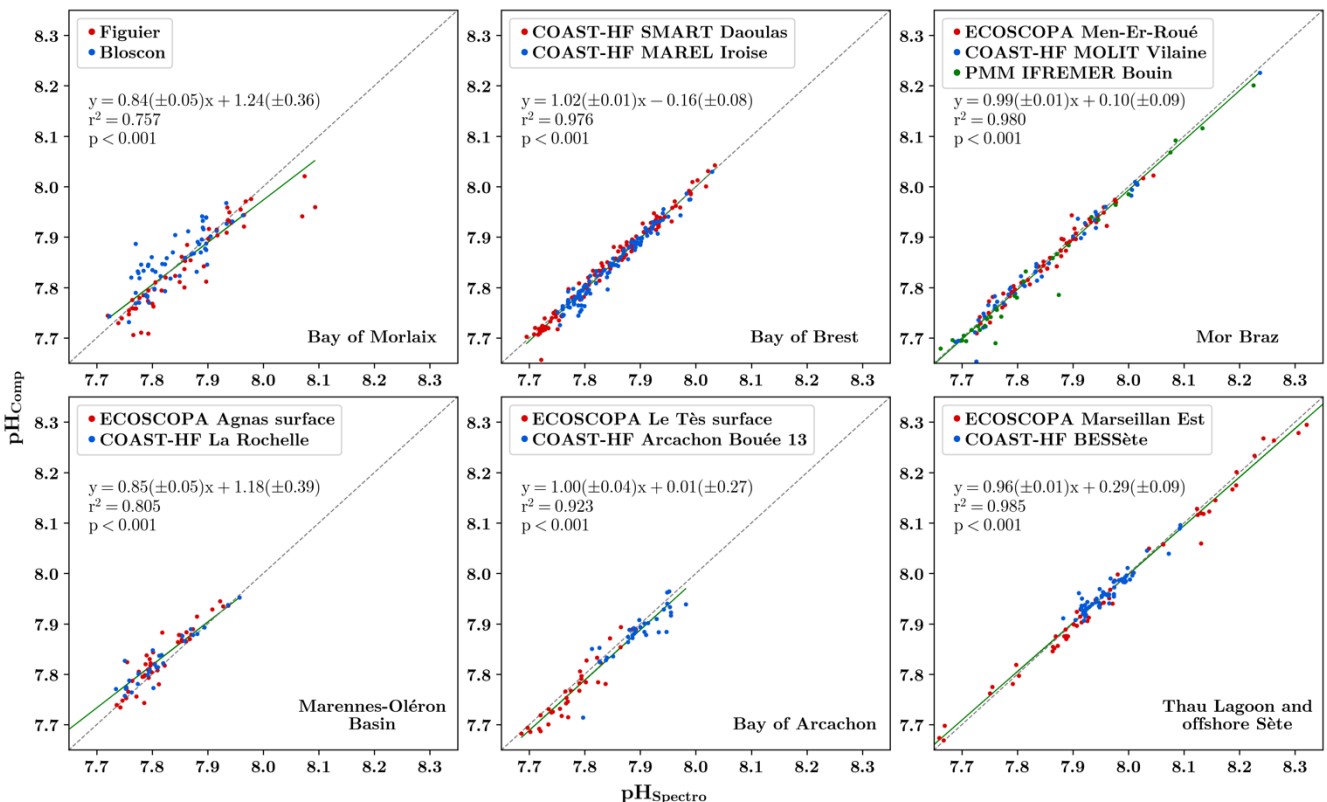

**Figure 3: Comparison of spectrophotometric pH$_T$ (pH on the total scale) measurements (pH$_{Spectro}$) with numerical pH estimation based on total alkalinity and dissolved inorganic carbon measurements (pH$_{Comp}$) in the six different regions (red dots refer to farmed sites and blue dots to oceanic sites). The obtained linear relationship is represented by a solid green line and the regression coefficient (r$^2$) is indicated and the p-value are indicated. The black dashed line represents the 1:1 relationship.**

### 3.1.3 High-frequency data quality

As prescribed by McLaughlin et al. (2017), the typical offset between spectrophotometric reference samples and the corrected SeaFET pH$_T$ time series must range between -0.02 and 0.02 pH$_T$ units. The corrected SeaFET data demonstrate strong agreement with the discrete spectrophotometric measurements, as illustrated in Fig. 4. The relationships between spectrophotometric pH$_T$ and SeaFET pH, both at *in situ* temperature and at 25 °C, are highly significant, exhibiting r$^2$ values greater than 0.99 (p-value < 0.001). Due to the complexity of high-frequency monitoring and the multiple possible biases (temperature and conductivity sensors, time synchronizing error, numerical uncertainties in correction method, biofouling development, etc.), the final processed pH$_T$ signal fulfills the weather quality objective with an uncertainty of 0.02.



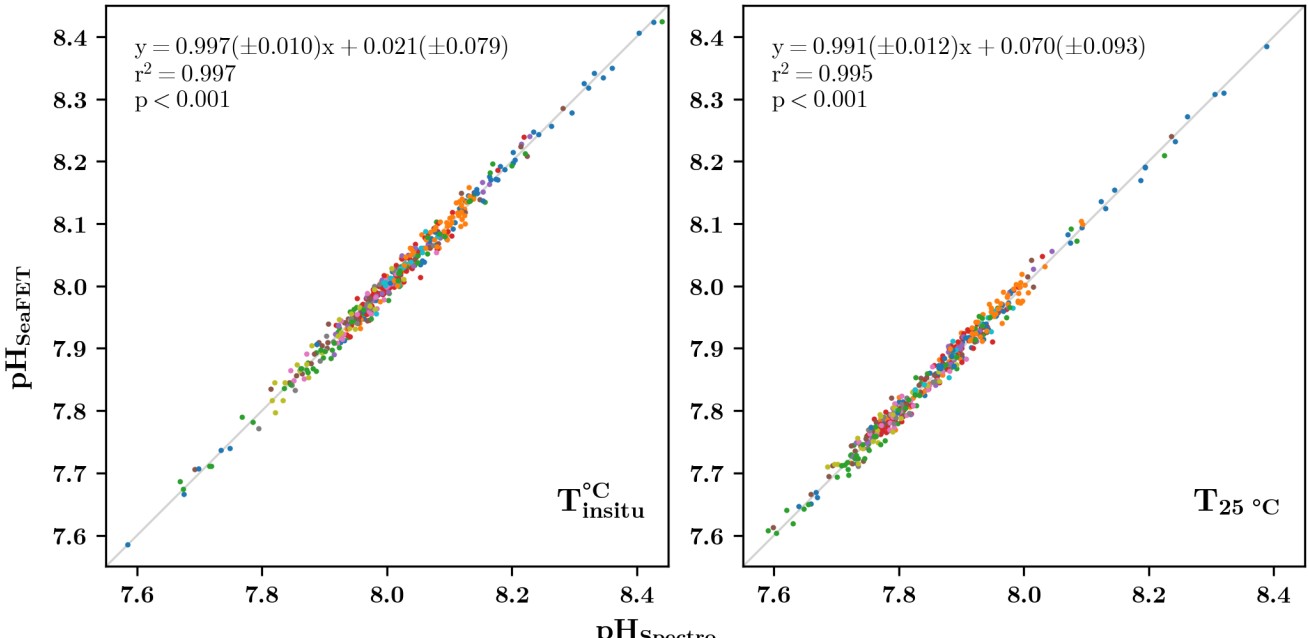

**Figure 4: Comparison of chosen $pH_T$ (pH on the total scale) corrected SeaFET signal ($pH_{SeaFET}$) versus spectrophotometric measurements ($pH_{Spectro}$) at *in situ* temperature (left) and at 25 °C (right). The color represents a unique site and the grey line represents the 1:1 relationship.**

## 410   3.2 Low-frequency total alkalinity and dissolved inorganic carbon variations in the different study sites

$A_T$ and $C_T$ data from discrete samples analyzed at the SNAPO-$CO_2$ analytical platform are shown in Fig. 5 (n = 641). According to the region, they exhibit various levels and seasonal ranges. Atlantic offshore sites present roughly similar $A_T$ and $C_T$ values with an overall means of 2322 ± 330 and 2127 ± 315 µmol kg$^{-1}$. The largest seasonal variations for Atlantic offshore sites are observed on the COAST-HF MOLIT Vilaine site (Mor Braz) as a consequence of the seasonal strong runoff from the Vilaine
and Loire rivers at this site (salinity range of 15). Among offshore areas, the Mediterranean site presents the highest level of $A_T$ and $C_T$ (2581 ± 81 and 2288 ± 87 µmol kg$^{-1}$, respectively) as a consequence of higher salinity levels than in the Atlantic (37.2 ± 1.5 vs 33.6 ± 1.5). With the exception of the Mor Braz area (i.e. strong influence of the Vilaine and Loire rivers on the salinity of the offshore site), farming sites generally present lower levels of both $A_T$ and $C_T$ compared to oceanic sites, generally related to lower salinity levels.
Regarding seasonal variations, the highest ranges in both $A_T$ and $C_T$ are found in the most enclosed areas of the network, the Bay of Arcachon and the Thau lagoon. While the influence of freshwater runoff explains the large ranges of salinity as well as $A_T$ and $C_T$ changes in the Bay of Arcachon, the situation is opposite in the Thau lagoon where summer evaporation engenders the maximum salinity of 41, driving strong increases in both $A_T$ and $C_T$ in this area. It is worth noting that larger ranges in $C_T$ than in $A_T$ in the Thau lagoon suggest a significant additional impact of biology (photosynthesis/respiration) on carbonate
chemistry (see 3.3).





The terrestrial Bouin site also presents among the highest values of the dataset with large seasonal variations both for $A_T$ and $C_T$ (2367 ± 489 and 2182 ± 472 µmol kg$^{-1}$, respectively). The calcium concentration observed in the soil composition in Vendée (Hájek et al., 2021) is a plausible explanation for these data patterns.



**Figure 5: Distribution of total alkalinity ($A_T$) and dissolved inorganic carbon ($C_T$) concentrations obtained from discrete sampling in farming (red) and oceanic sites (blue). The Ifremer Bouin oyster growth facility is shown in green. The black dashed horizontal line corresponds to the average value of the whole dataset ($\overline{A_T}$ = 2355.7 µmol kg$^{-1}$ and $\overline{C_T}$ = 2147.0 µmol kg$^{-1}$)**

### 3.3 High-frequency pH$_T$ variations in the different study sites

High-frequency pH$_T$ measurements together with discrete (spectrophotometric) pH$_T$ measurements acquired from January
2021 to December 2022 at all sites are shown in Fig. 6. Offering an insightful statistical summary, Table 3 provides a site-



specific analysis based on data acquired during this period. The differences observed among regions can be attributed to their unique hydrological and geographical characteristics. The highest average $pH_T$ levels are measured on both sites (farming and offshore) located in the Mediterranean Sea ($pH_T \sim 8.09$) while the lowest is measured inside the Bay of Arcachon, presenting the highest difference in mean $pH_T$ between the farming and the oceanic sites in our network ($pH_T$ 7.93 *vs* 8.01, respectively).

At the exception of the Bay of Arcachon, farming and oceanic sites do not present strong differences in their overall mean $pH_T$ level but farming sites generally present a larger seasonal amplitude (Table 3). The largest differences in seasonal amplitude between farming and oceanic sites are observed in the Mor Braz, the Bay of Arcachon and the offshore Sète in the Mediterranean area. It was already mentioned that the oceanic site in the Mor Braz is much more influenced by the Vilaine and Loire rivers than the farming site, resulting in overall lower salinity and $pH_T$ levels as well as stronger seasonal changes in these variables due to freshwater runoff. In contrast, in the Bay of Arcachon, lower levels and amplitude of $pH_T$ variations are observed inside the Bay at the farming site. The Thau lagoon is by far the site experiencing the largest seasonal $pH_T$ variations, with a maximum reached in April ($pH_T \sim 8.51$) and a minimum in September ($pH_T \sim 7.45$). In this region, differences between the farming and offshore sites can be explained by the location of the farming site within the Thau Lagoon, exhibiting weak eutrophic to oligotrophic conditions, in contrast to the BESSète site subjected to oligotrophic conditions in the

Mediterranean offshore waters although transient changes in $pH_T$ can be observed as a consequence of the influence from the Rhone and Herault rivers. The Bouin facility undergoes two different annual cycles shaped by the specific demands of the nursery. Water introduced through pumping into various basins, experiences renewal time within different periods ranging from days to months, contingent on hydrological watershed conditions. In all sites, $pH_T$ normalized at 25°C ($pH_T^{25°C}$; data not shown) reveals that most of the observed seasonal variations are independent of temperature but more likely reflect changes

in water masses and in the biological balance between autotrophy and heterotrophy.

Farming sites generally show more pronounced daily variations, which include both tidal and diurnal variations. This phenomenon is attributed to stronger terrestrial runoff impacts and/or more intense biological activities in these locations. It is interesting to note that regions such as the Bay of Morlaix, the Bay of Brest, and the Marennes-Oléron Basin, characterized by low freshwater runoff, display comparable daily average $pH_T$ levels between the oceanic and farming sites, except during

summer when biological activities are more intense. For instance, the farming site Figuier (Bay of Morlaix) displays a maximum diurnal variation of 0.15 $pH_T$ in July, a value two times larger than that observed at the offshore Bloscon site (0.07 $pH_T$). Similarly, the Bay of Brest shows greater diurnal variations during the biologically productive periods, particularly from spring to autumn. The farming site SMART Daoulas shows an average daily $pH_T$ variation of 0.11 unit during this period, while the offshore MAREL Iroise site records a lower maximum variation of 0.06 $pH_T$ during the springtime. In the

Mediterranean Sea, diurnal variations within the lagoon remain relatively modest, peaking only at 0.09 in September, just after intense phytoplankton bloom events. Again, the Mor Braz case stands out due to the influence of the Loire River, which brings large amounts of nutrients and thus induces phytoplankton blooms, at the oceanic site MOLIT Vilaine. It results in significant $pH_T$ variations, particularly noticeable over 15-day periods in July 2021 and June 2022, reaching up to 0.14 unit. Conversely,





the farming site in the Bay of Quiberon displays varying hydrological conditions depending on climatic patterns, with major

diurnal variations reaching up to 0.08 $pH_T$ in summer. In the Bay of Arcachon, the disparities between the farming and offshore

sites can be attributed to the existing salinity gradient from the coast to the offshore, alongside a noticeable dependency on the

semi-diurnal tidal cycle. The average $pH_T$ at the farming site consistently remains slightly lower than the offshore counterpart

(-0.08 $pH_T$), with the most significant diurnal variations recorded across the network, reaching 0.17 $pH_T$ during biologically

productive periods. Finally, as a consequence of the longer water renewal time, at the Bouin hatching facilities, the diel

variations in $pH_T$ are high all year round with a peak in August during the biologically productive period (0.15 $pH_T$).

**Figure 6: High-frequency pH$_T$ (pH on the total scale) time series in farming (red) and oceanic sites (blue) at *in situ* temperature. The larger red and blue dots show discrete pH$_T$ values obtained using a spectrophotometric method in farming and oceanic sites, respectively. The lower plot (in green) shows the pH$_T$ evolutions inside the Bouin oyster growth facility at Ifremer**



**Table 3: General statistics for each site based on pH$_T$ (pH on the total scale) data acquired between January 2021 and December 2022.**

| Site | Minimum | Q10 | Mean<br>$\sigma$ | Q90 | Maximum |
|---|---|---|---|---|---|
| **Bay of Morlaix** | | | | | |
| Figuier | 7.75 | 7.93 | **8.03**<br>*0.09* | 8.17 | 8.38 |
| Bloscon | 7.86 | 7.95 | **8.02**<br>*0.06* | 8.1 | 8.26 |
| **Bay of Brest** | | | | | |
| COAST-HF SMART Daoulas | 7.66 | 7.89 | **7.99**<br>*0.09* | 8.13 | 8.28 |
| COAST-HF MAREL Iroise | 7.86 | 7.97 | **8.02**<br>*0.06* | 8.08 | 8.26 |
| **Mor Braz** | | | | | |
| ECOSCOPA Men-Er-Roué | 7.8 | 7.93 | **8.02**<br>0.08 | 8.13 | 8.3 |
| COAST-HF MOLIT Vilaine | 7.62 | 7.87 | **8.00**<br>*0.11* | 8.17 | 8.49 |
| **Marennes Oléron Basin** | | | | | |
| ECOSCOPA D'Agnas | 7.72 | 7.88 | **7.96**<br>*0.06* | 8.03 | 8.16 |
| La Rochelle | 7.73 | 7.85 | **7.96**<br>*0.08* | 8.04 | 8.26 |
| **Bay of Arcachon** | | | | | |
| ECOSCOPA Le Tès | 7.47 | 7.83 | **7.93**<br>*0.07* | 8.02 | 8.1 |
| COAST-HF Bouée 13 | 7.84 | 7.95 | **8.01**<br>*0.05* | 8.07 | 8.18 |
| **Thau Lagoon and offshore Sète** | | | | | |
| ECOSCOPA Marseillan | 7.45 | 7.74 | **8.09**<br>*0.22* | 8.35 | 8.51 |
| COAST-HF BESSète | 7.87 | 8.02 | **8.09**<br>*0.04* | 8.13 | 8.3 |
| Ifremer Bouin oyster growth facility | 7.43 | 7.74 | **7.96**<br>*0.19* | 8.21 | 8.63 |

### 3.4 High-frequency assessment of carbonate chemistry in the different sites

The determination of the carbonate system at high frequency is achieved based on high-frequency pH$_T$, reconstructed high-frequency $A_T$ data (see §2.3.4) as well as temperature, salinity and nutrient data. Results of the multiple regression analyses allowing the reconstruction of high-frequency $A_T$ data based on temperature, salinity, ammonium, phosphorus and silicate are presented in Table 4. For both sites located in the Bay of Morlaix, no significant linear relationship was found with salinity. This is most certainly due to the data covering a small salinity range at these sites (33.0 < salinity < 35.3). Gac et al. (2020) analyzed several seasonal surveys performed in this Bay, covering a much larger salinity gradient. They found a robust linear



relationship between $A_T$ and salinity ($A_T = 50.4 * S + 575$, n = 236, $r^2 = 0.98$), which is then used in our study. For other sites,
salinity is the major driver with a positive effect except for the Marennes-Oléron basin. Depending on the site, the relationship takes into account nutrient data, which indicates generally runoff impacts. ECOSCOPA farming sites in Bay of Arcachon and Bay of Brest are typical examples with $r^2$ values greater than 0.8 (p < 0.001). Overall, simulated $A_T$ was strongly correlated with measured values ($r^2 = 0.93$, p < 0.001, n = 641).

**Table 4: Summary of stepwise multiple regression analyses using seawater temperature, salinity as well as ammonium, phosphorus**
**and silicate concentrations as explanatory variables and alkalinity as a response variable for each site.**

| Site | Step | Explanatory variables | Parameter estimate | BIC | $\sum r^2$ | *P* value |
|---|---|---|---|---|---|---|
| Figuier | 0 | Intercept | 2320.99 | 364.94 | 0.0000 | <0.0001 |
| Bloscon | 0 | Intercept | 2343.98 | 339.32 | 0.0000 | <0.0001 |
| COAST-HF SMART Daoulas | 0 | Intercept | 237.10 | 848.14 | 0,0000 | 1.0000 |
|  | 1 | Salinity | 63.52 | 720.76 | 0.7161 | <0.0001 |
|  | 2 | Temperature | -8.40 | 691.40 | 0.7906 | <0.0001 |
|  | 3 | Phosphorus | -57.16 | 677.32 | 0.8210 | <0.0001 |
| COAST-HF MAREL Iroise | 0 | Intercept | 666.43 | 302.86 | 0,0000 | 1.0000 |
|  | 1 | Salinity | 49.81 | 274.19 | 0.5430 | 0.0000 |
|  | 2 | Phosphorus | -129.58 | 264.03 | 0.6671 | 0.0005 |
|  | 3 | Temperature | -4.99 | 261.98 | 0.6992 | 0.0345 |
| ECOSCOPA Men-Er-Roué | 0 | Intercept | 1582.33 | 411.33 | 0,0000 | 1.0000 |
|  | 1 | Salinity | 25.08 | 381.55 | 0.4437 | 0.0000 |
|  | 2 | Temperature | -6.65 | 379.95 | 0.4692 | 0.0672 |
|  | 3 | Phosphorus | -87.83 | 371.51 | 0.5657 | 0.0010 |
| COAST-HF MOLIT Vilaine | 0 | Intercept | 1225.14 | 458.57 | 0,0000 | 1.0000 |
|  | 1 | Salinity | 37.37 | 404.34 | 0.6979 | 0.0000 |
|  | 2 | Temperature | -7.98 | 404.07 | 0.7066 | 0.1314 |
|  | 3 | Phosphorus | -93.13 | 400.84 | 0.7368 | 0.0169 |
| ECOSCOPA D'Agnas | 0 | Intercept | 3109.17 | 342.15 | 0,0000 | 1.0000 |
|  | 1 | Salinity | -18.73 | 317.51 | 0.4728 | 0.0000 |
|  | 2 | Ammonium | -20.44 | 315.11 | 0.5193 | 0.0334 |
|  | 3 | Temperature | -3.17 | 314.57 | 0.5436 | 0.0875 |
| La Rochelle | 0 | Intercept | 3029.88 | 200.52 | 0,0000 | 1.0000 |
|  | 1 | Salinity | -18.20 | 187.76 | 0.4175 | 0.0001 |
| ECOSCOPA Le Tès | 0 | Intercept | 394.86 | 331.29 | 0,0000 | 1.0000 |
|  | 1 | Salinity | 57.93 | 267.41 | 0.8636 | 0.0000 |
|  | 2 | Temperature | -4.98 | 262.30 | 0.8900 | 0.0060 |
| COAST-HF Bouée 13 | 0 | Intercept | 686.24 | 247.17 | 0,0000 | 1.0000 |
|  | 1 | Salinity | 47.37 | 215.62 | 0.6321 | 0.0000 |



| | | | | | | |
|---|---|---|---|---|---|---|
| ECOSCOPA Marseillan | 0 | Intercept | 4967.59 | 506.45 | 0,0000 | 1.0000 |
| | 1 | Salinity | -61.36 | 440.47 | 0.7171 | <0.0001 |
| | 2 | Silicate | -1.58 | 440.32 | 0.7241 | 0.1324 |
| COAST-HF BESSète | 0 | Intercept | 2974.21 | 373.70 | 0,0000 | 1.0000 |
| | 1 | Salinity | -9.54 | 353.80 | 0.3185 | <0.0001 |
| | 2 | Temperature | -2.32 | 345.66 | 0.4295 | 0.0013 |
| Ifremer Bouin oyster growth facility | 0 | Intercept | 1675.41 | 332.80 | 0,0000 | 1.0000 |
| | 1 | Salinity | 23.19 | 314.26 | 0.4390 | 0.0000 |
| | 2 | Temperature | 5.51 | 313.46 | 0.4734 | <0.0001 |

Based on computations using high-frequency data (temperature, salinity and interpolated nutrients), the monthly distribution of seawater saturation states with respect to aragonite ($\Omega_{Aragonite}$) is presented for all sites in both farming and oceanic areas in Fig. 7. The annual cycle varies among Atlantic areas, with the maximum value being reached between early spring and late
autumn. Here again, the Thau lagoon exhibits a unique cycle, with highly favorable conditions in spring and extremely low saturation state values in autumn. To evaluate the risk for shellfish production, pie charts illustrate the ratio of time spent below 1 (*i.e.* corrosive seawater). For now, based on the limited database, there are only a few periods in winter in the Bay of Arcachon and next to the Loire River mouth where the availability of carbonate ions could be of concern.

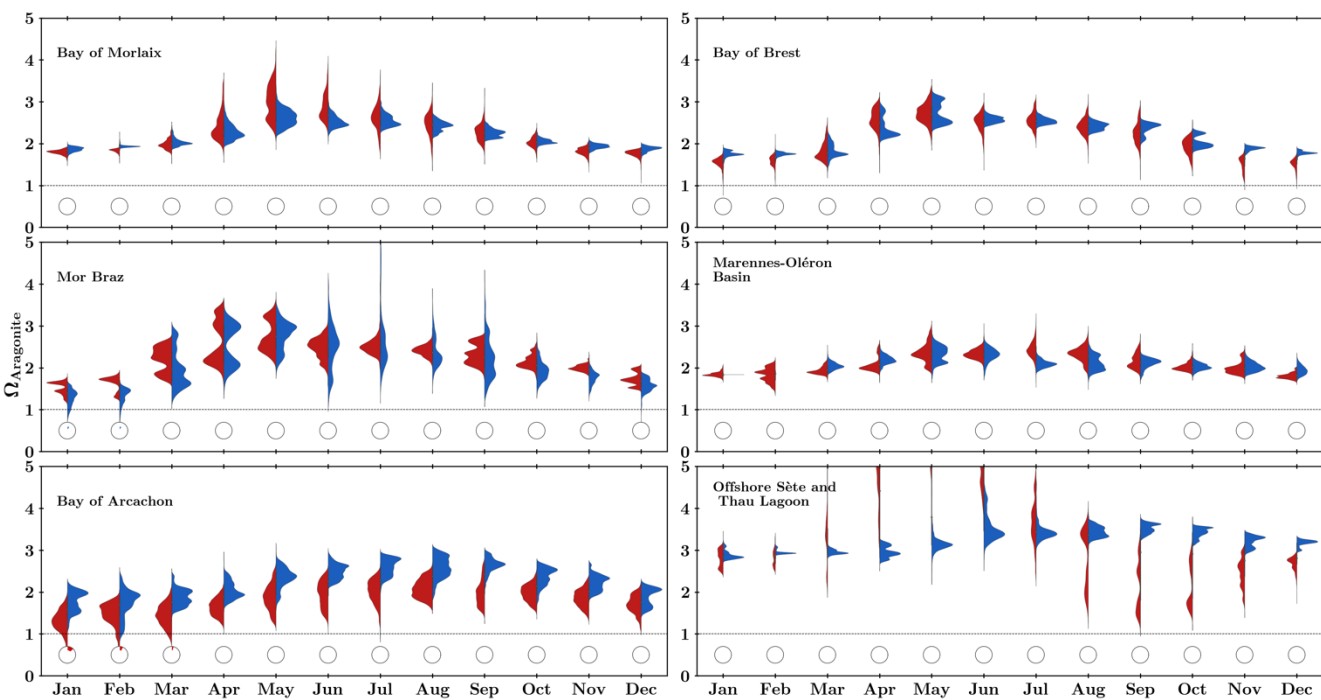


**Figure 7: Monthly distribution of seawater saturation state with respect to aragonite ($\Omega_{Aragonite}$) in the six study sites for farming (red) and oceanic (blue) areas. Pie charts represent the fraction of time spent below threshold 1 (represented as a horizontal dotted line).**

To fully understand when these acidification events occur, Fig. 8 presents the distribution of aragonite saturation state for

each region according to high-frequency temperature and salinity data. All Atlantic sites show a similar pattern with higher

saturation states during summer with acidification events only occurring during periods of freshwater runoff (salinity threshold

between 28 and 32 depending on the area). In contrast, acidification events in the Thau lagoon in the Mediterranean Sea are

correlated to over-salinity events at the end of summer.





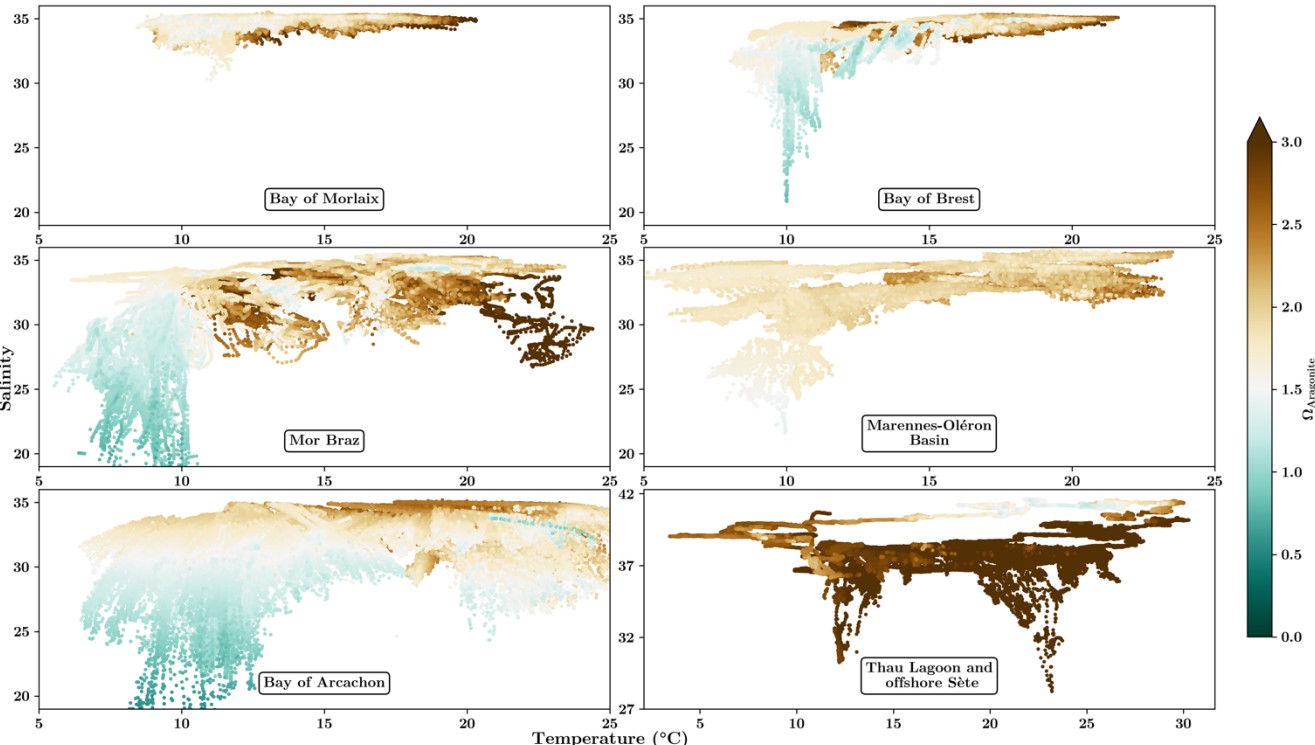

**Figure 8: Distribution of seawater saturation state with respect to aragonite ($\Omega_{Aragonite}$) for each region in relation to temperature and salinity**

## 4 Data availability

In order to facilitate file organization, each site is assigned an acronym in the list given in Table 5. External high-frequency temperature and salinity data are given in a csv file called CTD_{*site*}.csv containing raw data with an appropriate quality flag to filter validated data (flag 1: good data). Another csv file called pH_{*site*}.csv contains validated $pH_T$ data together with temperature and salinity (interpolated when necessary). The history deployment information is given in another file named deployment_{site}.csv including detailed maintenance operations and noticeable remarks concerning the SeaFET electrodes. Eventually, the discrete sample measurements are gathered in samples.csv with result from PyCO2SYS at *in situ* temperature and nutrient data from the SOMLIT and REPHY databases.

**Table 5: Output files acronym matching for the different sites**

| Site | Acronym |
|------|---------|
| Figuier | FIGUIER |
| Bloscon | BLOSCON |
| COAST-HF SMART Daoulas | RO16 |





| | |
|---|---|
| COAST-HF MAREL Iroise | MAREL |
| ECOSCOPA Men-Er-Roué | QB02 |
| COAST-HF MOLIT Vilaine | MOLIT |
| ECOSCOPA D'Agnas | MA03 |
| La Rochelle | ROCHL |
| ECOSCOPA Le Tès | AR11 |
| COAST-HF Bouée 13 | ARCB13 |
| ECOSCOPA Marseillan | TH03 |
| COAST-HF BESSète | BESS |
| Ifremer Bouin oyster growth facility | BOUIN |

High-frequency data and discrete sampling data are available on this repository at https://doi.org/10.17882/96982 (Petton et al., 2023a). The data is being provided free of charge to the public and scientific community, with the expectation that its broad distribution will result in increased comprehension and the emergence of novel scientific perspectives.

## 530 5 Conclusion and present status of the network

The network initiated in 2021 along the French coast area has provided essential data for the assessment of carbonate chemistry dynamics at various temporal scales and in contrasted coastal sites (shellfish farms located close to the shore *vs* sites with no shellfish farming more subjected to oceanic conditions). The large number of sites and geographical coverage of the network has already allowed us to evaluate the influence of diverse physical, chemical and biological processes (freshwater inputs, 535 tides, temperature, biological processes) which we briefly presented in the present manuscript. The acquired dataset will undoubtedly be of great interest to the public and scientific communities in the future as our choice to base our network on existing monitoring activities not only allowed providing reliable data at a very high acquisition rate and at a lower financial cost, but also the possibility to rely on existing low frequency datasets (chlorophyll, nutrients, organic matter concentrations etc...) for assessing the inter-play between biology and the chemical environment. However, autonomous time-series 540 acquisition close to shellfish farming involves a number of challenges mostly related to intense biofouling pressure. It explained most invalidated data even when employing fortnightly manual cleaning protocols which will require in the future the development of active and efficient antifouling solutions that are currently in development (localize chlorination, adapted wiper) in the framework of our project. Furthermore, the SeaFET technology is relatively novel when compared to more conventional temperature or conductivity sensors. We encountered disparities within the whole set of probes acquired with 545 specific electrodes malfunctioning within just a few months of deployment, despite the manufacturer SeaBird indicating a minimum of one-year service life. Adding complexity, a further issue emerged from June 2022 onwards: the SeaFET

maintenance service has been suspended due to a lack of the DuraFET component. While the service will apparently resume by late 2023, this disruption has already resulted in temporal gaps within the time series data. Additionally, evaluation of new sensors is underway with the objective of obtaining the reliability and accuracy of the already collected data.

**Author contributions**

SP, FP and FG designed the network and wrote the first draft of the manuscript. All co-authors actively participated in the process of acquiring data and amended the manuscript.

**Competing interest**

The authors declare that the research was conducted in the absence of any commercial or financial relationships that could be
construed as a potential conflict of interest.

**Acknowledgements**

We would like to thank all the personnel involved in the various stages of the network, built on the existing architecture of the ECOSCOPA and REPHY networks, and the SNO SOMLIT and COAST-HF from the Research Infrastructure ILICO (https://www.ir-ilico.fr). We also thank SNAPO-CO$_2$ analytical platform for the different analyses realized during this project.

**Financial support**

This network was funded by the CocoriCO$_2$ project (European Maritime and Fisheries Fund, 2020-2023). It also benefited from a subsidy from the Adour-Garonne water agency.

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
