# Peer review of "French coastal network for carbonate system monitoring: The CocoriCO2 dataset"

_Earth System Science Data, 2023_

## Author Response (AR1)

We thank the reviewer for their thoughtful and thorough comments, which will improve the overall quality of this manuscript. Here we repeat reviewer's comments in bold and provide our response below in normal font.

**It would be interesting to integrate the presentation of the monitoring sites with the tidal ranges given for all the sites, e.g., also for MOLIT and Men-Er- Roué as well as for the Thau lagoon and offshore Sete.**
Thank you for this remark. The average tidal range will be added to the two missing regions (5m for the Mor Braz area and below 1m for the Mediterranean sites).

**The authors should revise the pH data set "Ph_RO16 ", which refers to the site COAST-HF SMART Daoulas   n. 3804 data with pH>15, of which 1780 are flagged "1".**
Thank you for bringing attention to the identified invalid data. A script error was responsible for selecting incorrect pH values during two months in 2020. The pH data above 15 will be removed from the dataset.

**In the CTD dataset "CTD-AR11" most of the salinity data between 0 and 2 are flagged 4 but temperature data are flagged 1. Very low salinity values usually occur when the conductivity sensor of the CTD is not in seawater, so the corresponding temperature should not be considered a valid measurement (as in the CTD files of the other monitoring sites).**
You are correct regarding the CTD behavior, but the situation here differs. In the Bay of Arcachon, the bio-fouling pressure can be characterized as severe. During the most productive period in summer, conductivity sensor may become covered by microorganisms in less than 15 days, leading to discrepancies in salinity values. Despite this, the temperature data remain reliable. Special care was taken to identify periods when the CTD probe was out of the water for maintenance purposes, during which temperature data are flagged with a value of 1.

**The authors could also consider including in the discussion a comparison with the similar work of Simpson E., Ianson D., Kohfeld K. E., Franco A. C., Covert P. A., Davelaar M., and Perreault Y. Variability and drivers of carbonate chemistry at shellfish aquaculture sites in the Salish Sea, British Columbia. https://doi.org/10.5194/egusphere-2023-1553**
Thank you for your insightful comment. Since our publication was a data paper, we did not go deep into the discussion process. However, we acknowledge the importance of comparing our network with similar studies globally.
Notably, Simpson *et al.* (2023) conducted a comprehensive evaluation of diel and seasonal variability in carbonate chemistry conditions on the Canadian Pacific coast. Their study goes further by estimating the contribution of various drivers to seasonal and diel changes in pH and $\Omega$. Additionally, we will include a citation to Fujii *et al.* (2023), who conducted continuous pH monitoring in the subarctic coasts of Japan. Their work highlights critical levels of acidification for Pacific oyster larvae.

Fujii, M., Hamanoue, R., Bernardo, L. P. C., Ono, T., Dazai, A., Oomoto, S., Wakita, M., and Tanaka, T.: Assessing impacts of coastal warming, acidification, and deoxygenation on Pacific oyster (*Crassostrea gigas*) farming: a case study in the Hinase area, Okayama Prefecture, and Shizugawa Bay, Miyagi Prefecture, Japan, Biogeosciences, 20, 4527–4549, https://doi.org/10.5194/bg-20-4527-2023, 2023.

**Based on the information given in chapters 2.3.4. and 3.4, it is not entirely clear to me how the high-frequency AT was reconstructed by a multilinear model based on (high frequency) temperature and salinity measurements and (low frequency) phosphorus (COAST-HF, SMART, Daoulas, COAST-HF, MAREL Iroise, ECOSCOPA, Men-Er Roué) or ammonium (ECOSCOPA D'Agnas) measurements. Could the authors show the temporal variations of high frequency AT reconstructed at the different sites? The reconstructed total alkalinity data cannot be found in the datasets available online. Since the paper refers to the carbonate system and not just pH, I think the AT data should also be made available.**
The reviewer makes a great point and we concur. Following Table 4, we will incorporate a figure (attached herewith) illustrating the temporal variation of high-frequency reconstructed $A_T$, including also discrete measured $A_T$ data from the SNAPO-$CO_2$ laboratory. Moreover, the calculated $A_T$ will be available in the online dataset in each pH_{site}.csv file with a flag set to 5 (estimated value).

**For clarity I suggest to add Latitude and longitude in the Table 1 under the column heading "Position", and add "start date", and "recording frequency" under the column title "pHT monitoring".**
Agreed.

**In the caption of Table 2 the authors could better specify the "additional information" contained in the table.**
We assume the reviewer is speaking about the Table 3 where no information was given for mean value in bold, standard deviation in italic, $10^{th}$ and $90^{th}$ quartiles in Q10 and Q90 columns. As the reviewer also suggested in the technical corrections, the caption will be completed.

**Some technical corrections or possible improvements are suggested below:**

**Figure 5.  Explain the box plot shown (bars, dots …).**
The Figure 5 caption will be completed by the additional sentence: "Each box displays the quartiles (25% and 75%) of the dataset, with the median indicated by an orange line. The whiskers extend to illustrate the remaining distribution, reaching 1.5 times the inter-quartile range, with the exceptions for points identified as outliers."

**Line 480. Table 3 caption. Indicate the meaning of Q10 and Q90.**
"Q10 and Q90 denote the $10^{th}$ and $90^{th}$ quantiles, respectively. In the Mean column, the average value is highlighted in bold, while the standard deviation is presented in italic. » will be added to the table 3 caption.

**Line 495. Table 4 caption. Indicate the meaning of "BIC".**
In the column heading, the abbreviation "BIC" will be substituted with its explicit meaning, "Bayesian Information Criteria".

**Figure 7. Pie charts too small, not easy to read, better if enlarged.**
The pie charts will be enlarged to enhance readability.

**Figure 8.  Change the caption as it is unclear. I suggest a simpler form: Distribution of aragonite saturation state (ΩAragonite) in seawater for each region as a function of temperature and salinity.**
Agreed.

**I agree with comments from reviewer 1 (Dr. Giani), who also gave a very nice overview of the paper, so I will not reiterate any of his comments, especially given my tardy submission of this review (with apologies). Overall, the manuscript is well written and has attractive and clear graphics throughout. I have a few substantive comments and quite a few minor suggestions for edits to improve clarity of both text and figures below.**

We appreciate the reviewer for their comprehensive and constructive comments. It will enhance the overall quality of this manuscript and provide a more detailed description of each file of the dataset by incorporating greater homogeneity towards international standards. Here we repeat reviewer's comments in bold and provide our response below in normal font.

**Substantive comments:**

**I am especially glad that the authors will add the estimated high-frequency AT values to the data set, as suggested by Dr. Giani. Please make sure that it's clear that the data are \*estimated\* AT values for the benefit of data synthesis projects (similar to SOCAT) that may seek to integrate your data and require differentiating between measured and estimated data (in my SOCAT example, we do not accept estimated values, just measured values). As more coastal data are integrated into data products like these, it will be critical for metadata to be sufficient to allow end users to clearly differentiate which parameters are and are not appropriate for inclusion.**

Yes, the estimated high-frequency AT values will be named explicitly to avoid confusion.

**Along those lines, with respect to the data files:**

- **Consider using column headers as recommended in Liqing Jiang's 2022 best practices paper in Frontiers in Marine Science; these were developed to encourage consistency across the ocean carbon and acidification communities.**
- **At least for me, none of the CSVs opened properly because it seems that semi-colons were used in place of commas to separate values.**
- **The metadata file is not at all detailed compared to what I am used to (e.g. requirements of SOCAT or NOAA's Ocean Carbon and Acidification Data System). Some of the needed detail is in the manuscript, but there's detail lacking on things like the laundry list of CTD probes in lines 233-234 (individual errors for T and S on the various models—later on, some basic info is given, but I'm not sure if this will suffice for all end users). This is the reason for the "fair" rating on completeness. I am aware that my expectations may be set by being in more of a "climate quality" ocean carbon observation community, so I will defer to the editor on whether the level of detail in the metadata is suitable for publication in ESSD.**

Thank you for bringing attention to the limited metadata detail provided in the dataset. Our primary focus has been on enhancing details within the manuscript rather than on the files themselves. Recognizing the objectives of the ESSD journal, we plan to add a more comprehensive header to each file. It will be based on the structure found in NOAA's Ocean Carbon and Acidification Data System. Uncertainties will be written for each parameter. Additionally, we acknowledge the necessity of adhering to Liqing Jiang's 2022 best practices for naming column headers. To enhance clarity, we will specify the scale for temperature and salinity data, utilizing ITS-90 and PSS78, respectively. We also apologize for the inconvenience caused by the original files' delimiter format, which stemmed from the standard export format of Python scripts using the French regional language. This will be addressed by using the comma delimiter. Furthermore, there was a confusion between flags used for high-frequency data, based on Ocean Data Standards (IOC, 2013), and the one used for low-frequency data, based on WOCE primary level quality control flags for discrete samples. To mitigate confusion, we will adopt a unified flagging system based on Jiang et al's (2022) recommendations. This system includes the following flags: 0 for interpolated or calculated data, 1 for not evaluated/quality unknown, 2 for acceptable, 3 for questionable, 4 for known bad, 6 for the median of replicates, and 9 for missing values. This approach will also be extended to appropriately identify estimated high-frequency Alkalinity, attributing them the flag 0 for consistency across the dataset.

**Figure 1—text within the figure is mostly quite small and could be enlarged**

Thank you for this suggestion, all the labels of Figure 1 will be enlarged.

**Line 372—The sentence starting with "results" is confusing because you say "below the uncertainty level of 0.15%"—but it sounds like you mean >0.15% because then you say the data are "weather" quality. You may need to add a few more words here to clarify.**

This was a confusing mistake. Estimations of uncertainty of $A_T$ / $C_T$ measurements are mainly based on repeatability plus additional verification to standard CRM. Unfortunately, among the analyzed series there are repeatability values above the 3 $\mu$mol/kg$^{-1}$ threshold; therefore the data meet the "weather" quality objective. We will rephrase as follows: "The maximum repeatability of $A_T$ and $C_T$ measurements from the SNAPO-CO$_2$, flagged as "2: acceptable", reaches the uncertainty level of 0.15%".

**Line 380—It feels like the last sentence in this paragraph is referring to the spectrophotometric pH measurements, no? I'm also not sure why the climate-quality AT-CT data yield weather-quality estimated parameters. Is it because of the lower quality T and S (and silicate?) data?**

No, actually we were talking about the estimated $pH_T$ based on the low-frequency $A_T$ / $C_T$ pair. It was also misleading due to the discussed error pointed out in the previous comment. For the sake of clarity, we propagate the uncertainties by excluding all other uncertainty factors (temperature, salinity, nutrients, and equilibrium constants): It appears that the average uncertainty of estimated pH is 0.0059, still surpassing the climate-quality objective. We will clarify the sentence to prevent confusion by changing the last sentence to : "Again, the propagated uncertainties for the estimated $pH_T$ meet the "weather" quality objective"

**Section 3.3—In this section, some wording changes may be helpful. Diel means over 24 hr cycle, vs. diurnal, meaning daytime. When you talk about diel pH variation, please define what you mean. I have been assuming range, but it seems like it also could be SD. (Oh, now I see that Table 3 suggests its meant to be SD, though with quantiles, don't people often use medians and 25-75% quantiles instead of means and SDs?)**

Many thanks. In order to make sure there is no misunderstanding , we have used the wording "daily" when we referred to averages over 24 h. This is always the case and therefore we do not use "diurnal" or "diel" when we refer to 24h averages. e.g. "Farming sites generally show more pronounced daily (24 h) variations, which also include tidal variations." Diurnal is still used when we refer to tides as this is the common wording for (semi)-diurnal tides. In this section, the different given variations, mentioned as "the maximum daily variation…", relate to range and not SD. The general statistics given in Table 3 are intended to offer readers a more comprehensive perspective on the high-frequency dataset. Given the two-year duration of the monitoring period, median and mean value of $pH_T$ are really close. The addition of SD allows for the evaluation of inter-site variability. While a more detailed table could have included daily SD, semi-diurnal SD for the Atlantic sites, and even statistics with monthly frequency data aggregation, we deemed it potentially redundant and less engaging for the reader. Both quantiles 10% and 90% selection was based on their relevance to stakeholders' threshold values.

**Also, I'm a bit confused by the discussion about renewal time in basins at the Bouin facility because it was stated that sensors are at the seawater inlets, so I assumed these measurements should more or less reflect very nearshore conditions, but the language is unclear about whether the measurements reflect conditions in the facility or in the nearshore environment.**

We should have been clearer about the Bouin facility. This shore-based station is located near numerous oyster hatcheries on a polder. The polder is connected to the sea by several inlets and is essentially a network of interconnected basins whose water levels depend on tide and watershed conditions. During low-water conditions, some basins are not sufficiently renewed during low tides. To address this constraint, the facility manages the water level in each basin through different bypass communications. This ensures a continuous water incoming flux into the facility, but sometimes seawater renewal only occurs after a few tidal cycles. To prevent the SeaFET from drying, it is installed in a tank situated at the entrance of the seawater pumping circulation. We will provide a clearer site description by adding this information.

**Finally, whenever you refer to biological activity, you seem to be referring to production and respiration. Have you also considered whether the influences calcification and dissolution are reflected in your data sets? One potentially very important application of data sets like these would be to look at the effect of calcification on local biogeochemical conditions. It seems like your paired sites may be too far apart to use for this purpose, but maybe not…?**

We do not mention production and respiration in the text but rather use the term biological activity, which we now clarify as to be :"This phenomenon is attributed to stronger terrestrial runoff impacts and/or more intense biological activities (i.e. net community production and/or net calcification) in these locations.". As per the referee comment on whether or not our dataset could help in deciphering the impacts of net organic production vs net calcification on carbonate chemistry in the farming vs non-farming sites. We believe that our datasets will definitely be of interest for that purpose. However, indeed the sampling sites are certainly too far apart to be able to do so only based solely on our dataset and this should be combined with more geographically constrained spatial coverages of carbonate chemistry on farming sites and at their vicinity together with a precise quantification of benthic and pelagic organic matter biomass and on the influence of freshwater run-off on pH and alkalinity. This goes definitely beyond the data description paper that we propose.

**Table 4—I'm not following why a few of the sites have certain parameters included. For example, both Men-Er-Roué and Vilaine have T shown as being included in the equations, but BIC doesn't decrease by 2 or more (I'm assuming this is similar to how AIC works in that sense), the $R^2$ values only marginally increase, and the P values increase substantially instead of decreasing. Same thing for silicate at Marseillan. I would think those parameters would be excluded.**

Stepwise regression can sometimes include variables that may not seem meaningful, especially if the increase in R-squared is minimal and the associated p-values are not significant. This situation often arises because stepwise methods focus on incremental improvements in R-squared rather than considering the overall picture of model fit or the significance of each variable.
Here are a few reasons why seemingly non-significant variables might be included:
1. Sequential addition/removal: Stepwise regression adds or removes variables sequentially based on specific criteria (like p-values or information criteria). This can lead to variables being added that marginally improve the fit according to the criteria but might not contribute meaningfully.
2. Collinearity: Sometimes, a variable might be included because it improves the fit slightly, even though it is highly correlated with other variables already in the model. This can inflate the importance of a variable that doesn't add much new information.
3. Sample size: With a large sample, even small effects might achieve statistical significance. Stepwise methods might include these variables, but their practical significance could be minimal.
4. Overfitting: Including too many variables, even if they don't add much explanatory power, can lead to overfitting. This can reduce the model's ability to generalize to new data.

Considering these issues, it is crucial to exercise caution with stepwise regression results. It might be beneficial to evaluate the model's performance using techniques beyond R-squared and p-values. Methods like cross-validation, assessing residuals, and comparing different models based on information criteria (AIC, BIC) can provide a more comprehensive understanding of model performance and variable importance (which we did here). Also, domain knowledge and theoretical relevance should guide variable selection.
We also consider alternatives to stepwise regression, such as regularization methods (e.g. LASSO), which can address issues of overfitting and variable selection more effectively. We however were not entirely satisfied with the output obtained with the LASSO method which brings inconsistent results. We therefore opted for the simplest and the most intuitive STEPWISE method.

**Figure 7—Cool figure—I like your overall design. However, with so little of the time being corrosive, I wonder if you have well known thresholds for some of the important shellfish species. In the US Pacific Northwest, some species (e.g., Mediterranean mussels in Waldbusser et al 2014 in Nature Climate Change) have thresholds above 1, so it might be more germane to shellfish farming operations in your region to use a meaningful biological threshold instead of just using the thermodynamic arag=1.0 threshold.**

The referee is absolutely right, the figure has been changed in order to highlight the periods when Ωaragonite is below 1.5. This value has been chosen as per the results obtained by Ries et al. (2011) on adults of several mollusk species and by Waldbusser et al. (2015) as proposed by the referee, on larvae of Pacific oysters and Mediterranean mussels (two important species cultivated in France). Ries et al. (2011) show that for adults, negative growth appears below 1.5 for several mollusk species (Periwinkle, Whelk and soft Clams) and Walbbusser et al. (2015) report on chronic effects of Ωaragonite < 2 and especially strong increases in abnormal larvae below 1.5. As such, we have decided to use a conservative threshold of 1.5.

Revised text: "To evaluate the risk for shellfish production, pie charts illustrate the ratio of time spent below 1.5. This threshold has been chosen based on experimental data acquired by Ries et al. (2011) and Waldbusser et al. (2015) reporting, respectively, negative growth for several adult mollusk species and strong impacts (lower size, abnormal shape) on oyster and mussel larvae below this value. For now, based on the limited database, there are only a few periods in winter in the Bay of Arcachon and next to the Loire River mouth where the availability of carbonate ions could be of concern."

**Lines 546-547—In the US, we recently heard that the needed shallow-water FET chips are NOT going to be produced again (unless something has changed in the last few months), \*just\* the deepwater FET components. Not sure if this is the latest information (and doesn't necessarily require edits/response unless you have more current info), but disappointing to say the least.**

Thank you for sharing the detailed update on the Sea-Bird's efforts to resume production of the sensing elements for Deep SeapHOx and Shallow SeaFET/SeapHOx. Your concerns about the impact on our network are valid given the uncertainties surrounding the gel-based DuraFETs. The last official information we have : *Sea-Bird is evaluating the ability of its supplier to deliver high quality, gel-based DuraFETs on time*. This is indeed a dramatic situation, several of our sensors are already waiting for annual service at Seabird facility. Nevertheless, we are actively exploring alternative solutions to overcome this issue: A new infrared technology seems to be promising with sensors like AquapHOx from PyroScience© or pH logger from RBR©.

**Minor comments**

Thank you for pointing out these different inconsistencies in the text or in the figures. We will incorporate each of them, excluding the writing of frequency or interval as we had already followed ESSD house standards, which recommend not hyphenating modifiers containing abbreviated units. Here we repeat some specific reviewer's questions in bold and provide our response below in normal font.

**181-182—I don't understand "characterized as the first place for oyster's larvae recruitment in France"—do you mean naturalized recruitment of a non-native oyster or first place to see recruitment in an annual cycle or …?**

The Bay is an important area for oyster farming and is distinguished as the major place for oyster's larvae recruitment in France in terms of larval quantity.

**261—I suggest "validation" instead of "discrete" to avoid potential confusion. Discrete is often used in the sense of bottle samples, though I understand why you used the word for the validation casts. (Or do you mean the "discrete" bottle samples described in the previous paragraph? If this, then maybe moving this sentence to the previous paragraph will fix that problem.)**

We were talking about the "discrete" bottle samples described in the previous paragraph so we will move the sentence to the previous paragraph as you suggested.

**285—What does "Chart Datum" mean? Is this mean lower low sea level?**

Actually, it is similar and corresponds approximately to the Lower Astronomical Tide (LAT). In France, it is slighter below the LAT for safety reasons, between 0 and 20 cm. LAT inscription will be added to the table.

**319—"both electrodes"—not sure if you mean both reference electrodes? Or if you deployed two FETS. I think probably you are referring to internal/external ref electrodes—please add a word or two to clarify.**

Yes I was referring to the difference in signal between internal and external reference electrodes

**371—I'm confused—the "2: correct" flag is inconsistent with the flagging described on lines 321-322**

Yes this is confusing indeed as this flag refers to SNAPO-CO$_2$ laboratory, which follows WOCE (1994) quality control flags for variables measured from discrete water samples. As explained earlier, we followed OceanSite flags (IOC 2013) for high-frequency data and it differs from WOCE. This is why we will use a unified flagging system based on Jiang et al's (2022) recommendations unified flagging. So this parameter will be flagged as "2: Acceptable".

WOCE Operations Manual (1994), WHP Office Report 90-1, WOCE Report N°.67/91, p52-53. Woods Hole, Mass., USA.

**391—The previous lines made me wonder about whether there was an organic alkalinity issue too.**

Indeed, we are currently exploring the feasibility of directly measuring $A_T$ by each laboratory immediately after sampling. However, this approach comes with the challenge of additional analysis time and ensuring a satisfactory level of uncertainty comparable to SNAPO-CO$_2$ measurements requires a substantial effort (intercomparison exercise, buying CRM bottles). Our forthcoming strategy involves making our best attempt to expedite sample shipment to SNAPO-CO$_2$. It should help us to detect the presence of organic alkalinity in case of disparities.

---

## Author Response (AR2)

We appreciate both reviewers for raising last minor technical comments. We have taking them into account and incorporated the suggested revisions, including rephrased specific sentences. Here we repeat reviewers' comments in bold and provide our response when needed below in normal font.

**Reviewer 1**
**I think that the authors have adequately addressed the reviewers' comments and that the ms is almost ready to be accepted for publication in the journal. I have only a few comments that should be considered before final acceptance.**
**L.480-481. The maximum repeatability of AT and CT measurements from the SNAPO-CO2, flagged as "2: acceptable", reaches the uncertainty level of 0.15%. I am not sure if you are assuming that your measurements have an assessment higher of 0.15% if as I suppose is not the case you should rephrase the statement.**
Yes, this was not clear in the text. The measurements have an assessment that remains below the uncertainty level 0.15%. It is under the 0.4% criteria so they meet the "weather" quality objectives but not the "climate" one (0.1%).

**L. 654. At first sight, it is important to consider the importance of valid salinity high-frequency data, which induces directly a lack of estimated At.**
**Change the caption in order to avoid the redundancy of similar terms.**
Ok we simplify the caption of Figure 7 in order to avoid redundancy of similar terms.

**L. 684-686. Figure 8: Monthly distribution of seawater saturation state with respect to aragonite (ΩAragonite) in the six study sites for farming 685 (red) and oceanic (blue) areas. Pie charts represent the fraction of time spent below threshold 1 (represented as a horizontal dotted line).**
**Since the threshold was set at 1.5 therefore it would be congruent if also the pie charts represented the time spent below the saturation state threshold of 1.5 instead of 1.**
Thank you for pointing out this mistake in the caption. The threshold used for the pie charts is indeed 1.5.

**Reviewer 2**
**I really appreciate the authors' detailed replies explaining the areas of the manuscript that were initially unclear to me. Line numbers in my comments below are from the ATC1 file. In particular, I think the changes made to the QC and metadata practices strengthen the paper and will benefit end users. I appreciated hearing what they are learning about the evolving pH sensor availability situation.**
**I wanted to clarify my comment about using the data set for calcification rates. I thought the sites would be too far apart for this use, as they confirmed, but I intended the comment to be more a "future analysis" topic, perhaps deserving a mention but not a full treatment in this ESSD paper.**
Thank you for the clarification regarding the potential use of the dataset for calcification rates. We understand your intention now, and we agree that it would be a valuable topic for future analysis.

**Just a few other minor replies:**
**L194-195: I suggest that this wording might be clearer: "The Bay is an important area for oyster farming, characterized by having the most abundant larval oyster larvae recruitment in France."**
Included, thank you for this suggestion.

**233-238: Thanks for adding these details—it's helpful for understanding the results.**

**423-425: Thank you for the explanation. I appreciate you taking the time to spell it out. I think I was just looking at the table too quickly and misinterpreted a few things. I'll leave to the editor whether the added text is necessary.**
We have decided to retain this explanation in the manuscript as we consider it valuable for readers to understand the choice of the stepwise method.

**673-682 and Fig 8: I really like these changes! Much more regionally relevant this way.**